# Optimistic Posterior Sampling for Reinforcement Learning with Few Samples and Tight Guarantees

**Daniil Tiapkin**
HSE University
dtyapkin@hse.ru

**Denis Belomestny**
Duisburg-Essen University, HSE University
denis.belomestny@uni-due.de

**Daniele Calandriello**
DeepMind
dcalandriello@deepmind.com

**Éric Moulines**
École Polytechnique
eric.moulines@polytechnique.edu

**Remi Munos**
DeepMind
munos@deepmind.com

**Alexey Naumov**
HSE University
anaumov@hse.ru

**Mark Rowland**
DeepMind
markrowland@deepmind.com

**Michal Valko**
DeepMind
valkom@deepmind.com

**Pierre Ménard**
ENS Lyon
pierre.menard@ens-lyon.fr

## Abstract

We consider reinforcement learning in an environment modeled by an episodic, finite, stage-dependent Markov decision process of horizon $H$ with $S$ states, and $A$ actions. The performance of an agent is measured by the regret after interacting with the environment for $T$ episodes. We propose an optimistic posterior sampling algorithm for reinforcement learning (OPSRL), a simple variant of posterior sampling that only needs a number of posterior samples logarithmic in $H$, $S$, $A$, and $T$ per state-action pair. For OPSRL we guarantee a high-probability regret bound of order at most $\widetilde{\mathcal{O}}(\sqrt{H^3 SAT})$ ignoring $\text{poly} \log(HSAT)$ terms. The key novel technical ingredient is a new sharp anti-concentration inequality for linear forms which may be of independent interest. Specifically, we extend the normal approximation-based lower bound for Beta distributions by Alfers and Dinges [1984] to Dirichlet distributions. Our bound matches the lower bound of order $\Omega(\sqrt{H^3 SAT})$, thereby answering the open problems raised by Agrawal and Jia [2017b] for the episodic setting.

## 1 Introduction

In reinforcement learning an agent interacts with an environment, whose underlying mechanism is unknown, by sequentially taking actions, receiving rewards, and transitioning to the next state [Sutton and Barto, 1998]. With the goal of maximizing the expected sum of the collected rewards, the agent must carefully balance between *exploring* in order to gather more information about the environment and *exploiting* the current knowledge to collect the rewards. In this paper, we are interested in solving this exploration-exploitation dilemma by injecting noise into the agent's decision-making process.

We model the environment as an episodic, finite, unknown Markov decision process (MDP) of horizon $H$, with $S$ states and $A$ actions. In particular, we consider the *stage-dependent* setting where the

36th Conference on Neural Information Processing Systems (NeurIPS 2022).

rewards and the transition probability distributions can vary within an episode. After $T$ episodes, the performance of an agent is measured through *regret* which is the difference between the cumulative reward the agent could have obtained by acting optimally and what the agent really obtained.

Jin et al. [2018] and Domingues et al. [2020] provide a problem-independent lower bound of order $\Omega(\sqrt{H^3SAT})$ for this setting; see also Azar et al. [2017] for a lower bound when the transitions are stage-independent.

One generic solution to the exploration-exploitation dilemma is the *principle of optimism in the face of uncertainty*. A simple way to implement this principle consists in building *upper confidence bound (UCB)* on the optimal Q-value function through the addition of *bonuses* to the rewards. This is done by either model-based algorithms [Azar et al., 2017, Dann et al., 2017, Zanette and Brunskill, 2019] or model-free algorithms [Jin et al., 2018, Zhang et al., 2020, Menard et al., 2021]; see also [Jaksch et al., 2010, Fruit et al., 2018, Talebi and Maillard, 2018] for the non-episodic setting. Notably, among others, both the upper confidence bound value iteration (UCBVI) of Azar et al. [2017] and the UCB-Advantage algorithm of Zhang et al. [2020] enjoys a problem-independent regret bound[1] of order[2] $\widetilde{\mathcal{O}}(\sqrt{H^3SAT})$ that matches the aforementioned lower bound for $T$ large enough and up to terms poly-logarithmic in $H, S, A, T$.

Another way is to implement the optimism by *injecting noise*. A typical example is the random least-square value iteration (RLSVI, Osband et al., 2016b, Russo, 2019) algorithm which at each episode computes new Q-values by noisy value iteration from an estimated model and then acts greedily with respect to them. In particular, a Gaussian noise is added to the reward before applying the Bellman operator to encourage exploration. Indeed, when the variance of the noise is carefully chosen, it allows to obtain optimistic Q-values with at least a fixed probability. Russo [2019] first proved a regret bound of order $\widetilde{\mathcal{O}}(H^2S^{3/2}\sqrt{AT})$ for RLSVI. Later, Xiong et al. [2021] obtained an optimal regret bound of order $\widetilde{\mathcal{O}}(\sqrt{H^3SAT})$ for a modified version of RLSVI where the variance of the injected Gaussian noise is scaled by a term similar to the Bernstein bonuses used in UCBVI. Note that the RLSVI was also successfully extended beyond the tabular case to settings with function approximation, e.g. see Ishfaq et al., 2021, Zanette et al., 2020.

Recently, Pacchiano et al. [2021] analyzed a version of RLSVI where the Gaussian noise is replaced by a bootstrap sample of *the past rewards* and added pseudo rewards in the same fashion as Kveton et al. [2019]. The algorithm proposed by Pacchiano et al. [2021], comes with a regret bound of order $\widetilde{\mathcal{O}}(H^2S\sqrt{AT})$.

By generalizing the Thompson sampling algorithm [Thompson, 1933] originally given for stochastic multi-armed bandit, Osband et al. [2013] propose a posterior sampling for reinforcement learning (PSRL). PSRL algorithm also relies on noise to drive exploration. The general idea behind it is to maintain a *surrogate Bayesian model* on the MDP, for instance, a Dirichlet posterior on the transition probability distribution if the rewards are known. At each episode, a new MDP is sampled (i.e., a transition probability for each state-action pair) according to the posterior distribution of the Bayesian model. Then, the agent acts optimally in this sampled MDP. As the posterior is not well concentrated in the unexplored region of the MDP, the probability that the Q-value of the sampled MDP is optimistic in this region is high. Therefore, the agent will be incentivized to explore. Although the original Thompson sampling is well-studied in the frequentist setting [Agrawal and Goyal, 2012, Kaufmann et al., 2012, Agrawal and Goyal, 2013, Zhang, 2022] and the Bayesian setting [Thompson, 1933, Russo and Roy, 2016, Russo and Van Roy, 2014], most of the analysis of PSRL only provide Bayesian regret bounds [Osband et al., 2013, Abbasi-Yadkori and Szepesvári, 2015, Osband et al., 2016b, Ouyang et al., 2017, Osband and Van Roy, 2017], i.e., when the true MDP is effectively sampled according to the prior of the surrogate Bayesian model. Despite this lack of guarantees, PSRL demonstrates competitive empirical performance in comparison to bonus-based algorithms [Osband et al., 2013, Osband and Van Roy, 2017]. Additionally, the exploration mechanism used by PSRL (and RLSVI) was successfully extended outside the tabular setting and used in deep RL environments [Osband et al., 2016a, 2018, 2019].

---

[1] We translate all the bounds to the *stage-dependent* setting by multiplying the regret bounds in the stage-independent setting by $\sqrt{H}$, see Jin et al. [2018].

[2] In the $\widetilde{\mathcal{O}}(\cdot)$ notation we ignore terms poly-log in $H, S, A, T$.

One exception to the above is the work of Agrawal and Jia [2017b] that studies PSRL from a *frequentist* perspective in the infinite-horizon, non-episodic average reward setting. In particular, they provide a regret bound[3] of order[4] $\widetilde{\mathcal{O}}(H^2 S\sqrt{AT})$ for an optimistic version of PSRL that we call SOS-OPS-RL since it switches between two types of sampling of the transitions: (1) *simple optimistic sampling*, when the number of observed transitions at a given state-action pair is too small. In this case, the sampled transition is a random mixture between the uniform distribution over the states and an empirical estimate of the true transition biased by some bonus-like terms; or if the number of observed transitions at a given state-action pair is large enough (2) *optimistic posterior sampling*, where $\widetilde{\mathcal{O}}(S)$ samples from an inflated Dirichlet posterior are used instead of one sample used in PSRL. Then, from these $\widetilde{\mathcal{O}}(S)$ sampled transition probabilities we select the most optimistic one i.e., the one leading to the largest optimal Q-value.

The key idea underpinning the analysis of SOS-OPS-RL, and PSRL-like algorithms in general, is to control the deviations of the Dirichlet posterior on the transition probability distributions. In particular, we need to show that the *posterior spreads enough to ensure optimism*. To this end, Agrawal and Jia [2017b] derive an anti-concentration bound for any fixed projection of a Dirichlet random vector. The latter result in turn relies upon an equivalent representation of a Dirichlet vector in terms of independent Beta random variables and an anti-concentration bound for the corresponding Beta distribution. However, this anti-concentration inequality is not uniformly tight, in particular its polynomial dependence on the number of states $S$ is suboptimal.

Agrawal and Jia [2017b] conclude with two open problems. The first question is whether one can reduce the number of posterior samples required per state-action pair from $\widetilde{\mathcal{O}}(S)$ to constant or logarithmic in $S$. The second asks if it is possible to obtain a near-optimal regret bound and in particular to improve the dependence on $S$. In this paper, we *answer both of them in the affirmative* in the episodic setting. Indeed, we propose optimistic posterior sampling algorithm for reinforcement learning (OPSRL) that only requires $\widetilde{\mathcal{O}}(1)$ samples from an inflated posterior while enjoying a near-optimal problem independent regret bound of order $\widetilde{\mathcal{O}}(\sqrt{H^3 SAT})$. OPSRL is a simple optimistic variant of PSRL which, in particular, does not rely at all on "simple" (bonus-based) optimistic sampling.

The essential ingredient for OPSRL's analysis is our *novel anti-concentration bound for the projections of a Dirichlet random vector* (Theorem 3.3). We base it on a tight Gaussian approximation for linear forms of a Dirichlet random vector. This latter approximation can be seen as a substantial generalization to Dirichlet distributions of the result obtained by Alfers and Dinges [1984] for the case of Beta distributions. We obtain this approximation through a refined non-asymptotic analysis of the integral representation for the density of a linear form of a Dirichlet random vector, which was first derived[5] by Tiapkin et al. [2022]. We believe that the new anti-concentration inequality presented in this work could be of independent interest, e.g., to tighten or simplify analysis of non-parametric Thompson sampling like algorithms [Riou and Honda, 2020, Baudry et al., 2021a,b] for stochastic multi-armed bandits.

- We propose the OPSRL algorithm for tabular, stage-dependent, episodic RL. It is a simple optimistic variant of the PSRL algorithm that only needs $\widetilde{\mathcal{O}}(1)$ posterior samples per state-action pair. For OPSRL, we provide a regret bound of order $\widetilde{\mathcal{O}}(\sqrt{H^3 SAT})$ matching the problem independent lower bound up to poly-log terms. In particular we answer positively to two open questions by Agrawal and Jia [2017b] in the episodic setting.

- We derive a new anti-concentration inequality for a linear form of a Dirichlet random vector (Theorem 3.3) which is essential for the analysis of OPSRL. This result is a generalization to the Dirichlet case of the one provided by Alfers and Dinges [1984] for Beta distributions.

---

[3]As acknowledged by the authors, there was a mistake in the initial submission of their work where the previously announced bound was claimed to be $\sqrt{S}$ better, see Agrawal and Jia [2017a], Qian et al. [2020]

[4]We translate all the bounds from the infinite-horizon, non-episodic average reward setting to our setting by identifying the diameter with the horizon $H$ and multiplying the bound by $\sqrt{H}$ because of our stage-dependent transitions assumption.

[5]Note that the anti-concentration inequality proved by Tiapkin et al. [2022] based on the same integral representation is insufficient for our needs, see Remark 3.4 for a discussion.

## 2 Setting

We consider a finite episodic MDP $\left(\mathcal{S}, \mathcal{A}, H, \{p_h\}_{h\in[H]}, \{r_h\}_{h\in[H]}\right)$, where $\mathcal{S}$ is the set of states, $\mathcal{A}$ is the set of actions, $H$ is the number of steps in one episode, $p_h(s'|s,a)$ is the probability transition from state $s$ to state $s'$ by taking the action $a$ at step $h$, and $r_h(s,a) \in [0,1]$ is the bounded deterministic[6] reward received after taking the action $a$ in state $s$ at step $h$. Note that we consider the general case of rewards and transition functions that are possibly non-stationary, i.e., that are allowed to depend on the decision step $h$ in the episode. We denote by $S$ and $A$ the number of states and actions, respectively.

**Policy & value functions**  A *deterministic* policy $\pi$ is a collection of functions $\pi_h : \mathcal{S} \to \mathcal{A}$ for all $h \in [H]$, where every $\pi_h$ maps each state to a *single* action. The value functions of $\pi$, denoted by $V_h^\pi$, as well as the optimal value functions, denoted by $V_h^\star$ are given by the Bellman and the optimal Bellman equations,

$$Q_h^\pi(s,a) = r_h(s,a) + p_h V_{h+1}^\pi(s,a) \qquad V_h^\pi(s) = \pi_h Q_h^\pi(s)$$
$$Q_h^\star(s,a) = r_h(s,a) + p_h V_{h+1}^\star(s,a) \qquad V_h^\star(s) = \max_a Q_h^\star(s,a),$$

where by definition, $V_{H+1}^\star \triangleq V_{H+1}^\pi \triangleq 0$. Furthermore, $p_h f(s,a) \triangleq \mathbb{E}_{s' \sim p_h(\cdot|s,a)}[f(s')]$ denotes the expectation operator with respect to the transition probabilities $p_h$ and $\pi_h g(s) \triangleq g(s, \pi_h(s))$ denotes the composition with the policy $\pi$ at step $h$.

**Learning problem**  The agent, to which the transitions are *unknown* (the rewards are assumed to be known for simplicity), interacts with the environment during $T$ episodes of length $H$, with a *fixed* initial state $s_1$.[7] Before each episode $t$ the agent selects a policy $\pi^t$ based only on the past observed transitions up to episode $t - 1$. At each step $h \in [H]$ in episode $t$, the agent observes a state $s_h^t \in \mathcal{S}$, takes an action $\pi_h^t(s_h^t) = a_h^t \in \mathcal{A}$ and makes a transition to a new state $s_{h+1}^t$ according to the probability distribution $p_h(s_h^t, a_h^t)$ and receives a deterministic reward $r_h(s_h^t, a_h^t)$.

**Regret**  The quality of an agent is measured through its regret, that is the difference between what it could obtain (in expectation) by acting optimally and what it really gets,

$$\mathfrak{R}^T \triangleq \sum_{t=1}^{T} V_1^\star(s_1) - V_1^{\pi^t}(s_1).$$

**Counts**  The number of times the state action-pair $(s,a)$ was visited in step $h$ in the first $t$ episodes is denoted as $n_h^t(s,a) \triangleq \sum_{i=1}^{t} \mathbb{1}\{(s_h^i, a_h^i) = (s,a)\}$. Next, we define $n_h^t(s'|s,a) \triangleq \sum_{i=1}^{t} \mathbb{1}\{(s_h^i, a_h^i, s_{h+1}^i) = (s,a,s')\}$ the number of transitions from $s$ to $s'$ at step $h$.

**Improper Dirichlet distribution**  For $m \in \mathbb{N}^*$, the probability simplex of dimension $m$ is denoted by $\Delta_m$. For $\alpha \in (\mathbb{R}_{++})^{m+1}$, we denote by $\mathcal{D}ir(\alpha)$ the Dirichlet distribution on $\Delta_m$ with parameter $\alpha$. We also extend this distribution to improper parameter $\alpha \in (\mathbb{R}_+)^{m+1}$ such that $\sum_{i=0}^{m} \alpha_i > 0$ by injecting $\mathcal{D}ir((\alpha_i)_{i:\alpha_i>0})$ into $\Delta_m$. Precisely, we say that $p \sim \mathcal{D}ir(\alpha)$ if $(p_i)_{i:\alpha_i>0} \sim \mathcal{D}ir((\alpha_i)_{i:\alpha_i>0})$ and all other coordinates are zero.

**Additional notation**  For $N \in \mathbb{N}_{++}$, we define the set $[N] \triangleq \{1, \ldots, N\}$. We denote the uniform distribution over this set by $\mathcal{U}\text{nif}[N]$. The vector of dimension $N$ with all entries one is $\mathbf{1}^N \triangleq (1, \ldots, 1)^\intercal$. The empirical probability distribution $\widehat{p}_h^t(s,a)$ is defined as $\widehat{p}_h^t(s'|s,a) = n_h^t(s'|s,a)/n_h^t(s,a)$ if $n_h^t(s,a) > 0$ and $\widehat{p}_h^0(s'|s,a) = 1/S$ otherwise. Appendix A references all the notation used.

---

[6]We study deterministic rewards to simplify the proofs but our result extend to bounded random rewards as well.

[7]As explained by Fiechter [1994] and Kaufmann et al. [2020], if the first state is sampled randomly as $s_1 \sim p$, we can simply add an artificial first state $s_{1'}$ such that for any action $a$, the transition probability is defined as the distribution $p_{1'}(s_{1'}, a) \triangleq p$.

# 3 Algorithm

In this section we describe the OPSRL algorithm. In spirit, OPSRL proceeds similarly as PSRL except that it uses several posterior samples instead and acts optimistically with respect to them, explaining the name *Optimistic Posterior Sampling for Reinforcement Learning* (OPSRL).

**Optimistic pseudo-state** In order to define the prior used by OPSRL, we extend the state space $\mathcal{S}$ by an absorbing pseudo-state $s_0$ with reward $r_h(s_0, a) \triangleq r_0 > 1$ for all $h, a$ and transition probability distribution $p_h(s'|s_0, a) \triangleq \mathbb{1}\{s' = s_0\}$. A similar pseudo-state was already introduce in previous works, see for example Brafman and Tennenholtz [2002], Szita and Lőrincz [2008]. We denote by $\mathcal{S}' = \mathcal{S} \cup \{s_0\}$ the augmented states space and by $\Delta_{\mathcal{S}'}$ the set of probability distributions over $\mathcal{S}'$.

**Pseudo-counts** We define the pseudo-counts, $\overline{n}_h^t(s, a) \triangleq n_h^t(s, a) + n_0$, as the counts shifted by an initial value $n_0$. This shift corresponds to prior transitions to the pseudo-state, that is $\overline{n}_h^t(s'|s, a) \triangleq n_h^t(s'|s, a) + n_0 \mathbb{1}\{s' = s_0\}$. Similar to the empirical transitions, we define a pseudo-empirical transition probability distribution as $\overline{p}_h^t(s, a) = \overline{n}_h^t(s'|s, a)/\overline{n}_h^t(s, a)$.

**Inflated Bayesian model** Like PSRL, we define a Bayesian model on the transition probability distributions, except that the prior/posterior is inflated. The practice of inflating the posterior is common in the analysis of Thompson sampling like algorithm, see Agrawal and Jia [2017b], Abeille and Lazaric [2017]. Precisely, the inflated prior is a Dirichlet distribution $\mathcal{D}ir\left(\left(\overline{n}_h^0(s'|s, a)/\kappa\right)_{s' \in \mathcal{S}'}\right)$ parameterized by the initial pseudo-counts, and some constant $\kappa > 0$ controlling the inflation. Thus the prior is a Dirac distribution at a deterministic transition leading to the artificial state $s_0$. Then the inflated posterior is also a Dirichlet distribution $\mathcal{D}ir\left(\left(\overline{n}_h^t(s'|s, a)/\kappa\right)_{s' \in \mathcal{S}'}\right)$. Note that the prior is a proper prior (i.e., a valid probability distribution), but it will be updated in an improper way, i.e., probability transitions with no mass under the prior could get mass in the posterior, as they get positive counts.

**Optimistic posterior sampling** After episode $t$, for each state-action pair $(s, a)$ and step $h \in [H]$ we sample $J$ independent transition probability distributions $\widetilde{p}_h^{t,j}(s, a) \sim \mathcal{D}ir\left(\left(\overline{n}_h^t(s'|s, a)/\kappa\right)_{s' \in \mathcal{S}'}\right)$ from the inflated posterior. Then, the Q-values are obtained by optimistic backward induction with these transitions. Precisely the value after the last step is zero $\overline{V}_{H+1}^t(s) \triangleq 0$ and the optimal Bellman equations become

$$
\begin{aligned}
\overline{Q}_h^t(s, a) &\triangleq r_h(s, a) + \max_{j \in [J]} \widetilde{p}_h^{t,j} \overline{V}_{h+1}^t(s, a), \\
\overline{V}_h^t(s) &\triangleq \max_{a \in \mathcal{A}} \overline{Q}_h^t(s, a).
\end{aligned}
\tag{1}
$$

The next policy is greedy with the Q-values $\pi_h^{t+1}(s) \in \arg\max_{a \in \mathcal{A}} \overline{Q}_h^t(s, a)$. The complete procedure of OPSRL is described in Algorithm 1 for a general family of distributions parameterized by the pseudo-counts over the transitions instead of the inflated Dirichlet prior/posterior.

## 3.1 Analysis

We fix $\delta \in (0, 1)$ and the number of samples

$$
J \triangleq \lceil c_J \cdot \log(2SAHT/\delta) \rceil,
$$

where $c_J = 1/\log(2/(1 + \Phi(1)))$ and $\Phi(\cdot)$ is the cumulative distribution function (CDF) of a normal distribution. Note that $J$ has a logarithmic dependence on $S, A, H, T$, and $1/\delta$.

We now state the regret bound of OPSRL with a full proof in Appendix B. and a sketch in Section 3.2.

**Theorem 3.1.** *Consider a parameter $\delta \in (0, 1)$. Let $\kappa \triangleq 2(\log(12SAH/\delta) + 3\log(e\pi(2T + 1)))$, $n_0 \triangleq \lceil \kappa(c_0 + \log_{17/16}(T)) \rceil$, $r_0 \triangleq 2$, where $c_0$ is an absolute constant defined in (4); see Appendix B.2. Then for OPSRL, with probability at least $1 - \delta$,*

$$
\mathfrak{R}^T = \mathcal{O}\left(\sqrt{H^3 SATL^3} + H^3 S^2 AL^3\right),
$$

---
**Algorithm 1** OPSRL
---

1: **Input:** Family of probability distributions $\rho : \mathbb{N}_+^{S+1} \to \Delta_{\mathcal{S}'}$ over transitions, initial pseudo-count $\overline{n}_h^0$, number of posterior samples $J$.

2: **for** $t \in [T]$ **do**

3:     For all $(s,a,h) \in \mathcal{S} \times \mathcal{A} \times [H]$, sample $J$ independent transitions

$$\widetilde{p}_h^{t-1,j}(s,a) \sim \rho\big(\overline{n}_h^{t-1}(s'|s,a)_{s' \in \mathcal{S}'}\big), \quad j \in [J].$$

4:     Optimistic backward induction: set $\overline{V}_{H+1}^{t-1}(s) = 0$ and recursively for $h \in [H]$, compute

$$\overline{Q}_h^{t-1}(s,a) = r_h(s,a) + \max_{j \in [J]}\big\{\widetilde{p}_h^{t-1,j}\overline{V}_{h+1}^{t-1}(s,a)\big\},$$

$$\overline{V}_h^{t-1}(s) = \max_{a \in \mathcal{A}} \overline{Q}_h^{t-1}(s,a),$$

$$\pi_h^t(s) \in \arg\max_{a \in \mathcal{A}} \overline{Q}_h^{t-1}(s,a).$$

5:     **for** $h \in [H]$ **do**
6:         Play $a_h^t = \pi_h^t(s_h^t)$.
7:         Observe $s_{h+1}^t \sim p_h(s_h^t, a_h^t)$.
8:         Increment the pseudo-count $\overline{n}_h^t(s_{h+1}^t|s_h^t, a_h^t)$.
9:     **end for**
10: **end for**

---

*where $L \triangleq \mathcal{O}(\log(HSAT/\delta))$.*

**Computational complexity**   OPSRL is a model-based algorithm, and thus gets the $\mathcal{O}(HS^2A)$ space complexity as PSRL. Since we need $\widetilde{\mathcal{O}}(1)$ posterior samples per state-action pair the time complexity of OPSRL is of order $\widetilde{\mathcal{O}}(HS^2A)$ per episode, the same as PSRL up to poly-logarithmic terms. Building on the idea of Efroni et al. [2019], in Appendix F we propose the Lazy-OPSRL algorithm a more time-efficient version of OPSRL. Instead of recomputing the Q-value by backward induction before each episode, Lazy-OPSRL only performs one step of optimistic incremental planning at the visited states. It enjoys a regret bound of the same order $\widetilde{\mathcal{O}}(\sqrt{H^3SAT})$ as OPSRL but with an improved time-complexity per episode of $\mathcal{O}(HSA)$, see Theorem F.1 in Appendix F.

**Comparison with SOS-OPS-RL and PSRL**   One structural difference between OPSRL and SOS-OPS-RL of Agrawal and Jia [2017a] is that OPSRL only relies on optimistic posterior sampling while SOS-OPS-RL also uses simple optimistic sampling: a mixture of the uniform distribution over the states and an empirical estimate of the true transition kernel biased by some bonus-like terms. In particular, OPSRL does not use bonus-like quantities which could lead to poor empirical performance [Osband and Van Roy, 2017]. Another important issue is the number of posterior samples. SOS-OPS-RL needs $\widetilde{\mathcal{O}}(S)$ posterior samples in order to obtain a regret bound of order $\widetilde{\mathcal{O}}(H^2S\sqrt{AT})$ whereas OPSRL needs *only* $\widetilde{\mathcal{O}}(1)$ samples *and* obtains a better regret bound. Note that if we choose the number of posterior samples as $J = 1$ in OPSRL we recover PSRL up to two technical differences: First, the posterior is inflated in order to increase its variance. This technical trick was already used by Agrawal and Jia [2017a] and allows to guarantee optimism with a small number of posterior samples, see Section 3.2. Second, OPSRL uses a particular prior which is a Dirac distribution at a deterministic transition towards an optimistic pseudo-state. This prior is needed to control the deviations of the (inflated) posterior, see Theorem D.2.

**Comparison with RLSVI**   Both OPSRL and RLSVI build on the same mechanism for exploration. RLSVI just adds an Gaussian noise to the Q-values whereas OPSRL injects the noise naturally via a random transition sampled from a Dirichlet distribution. As controlling the deviation of the Q-value obtained with additive Gaussian noise is not difficult, the analysis of RLSVI is relatively straightforward [Russo, 2019, Ishfaq et al., 2021]. On the contrary the analysis of OPSRL is much more involved, see Section 3.2. However, the benefit of optimistic posterior sampling in OPSRL is

that it adapts *automatically* to the variance of the estimates of the transitions which is central for a regret bound with an optimal dependence on the horizon $H$ [Azar et al., 2017]. Adapting to the variance with RLSVI is much more involved and artificial, see Xiong et al. [2021]. This is probably one reason why RLSVI performs empirically worse than PSRL [Osband et al., 2016a].

## 3.2 Proof sketch

The proof of Theorem 3.1 consists of three important steps. The first step is devoted to the approximation for tails of weighted sums of Dirichlet distribution and embodies the main technical contribution of the paper.

**Step 1. Exponential and Gaussian approximation for Dirichlet distribution** The first result generalizes Riou and Honda [2020] to Dirichlet distributions with real parameters. Let us first recall the definition of the minimum Kullback-Leibler divergence for $p \in \Delta_m$ where $m \in \mathbb{N}^+$, a function $f : \{0, \ldots, m\} \to [0, b]$ for some $b \in \mathbb{R}^+$ and $u \in \mathbb{R}$,

$$\mathcal{K}_{\inf}(p, u, f) \triangleq \inf\{\mathrm{KL}(p, q) : q \in \Delta_m, qf \geq u\},$$

where we recall that $pf \triangleq \mathbb{E}_{X \sim p} f(X)$. This quantity appears already in the analysis of non-parametric bounded multi-arm stochastic bandits, see Honda and Takemura [2010], Cappé et al. [2013]. As the Kullback-Leibler divergence, the minimum Kullback-Leibler divergence admits a variational formula by Lemma 18 of Garivier et al. [2018] up to rescaling for any $u \in (0, b)$,

$$\mathcal{K}_{\inf}(p, u, f) = \max_{\lambda \in [0, 1/(b-u)]} \mathbb{E}_{X \sim p}[\log(1 - \lambda(f(X) - u))]. \tag{2}$$

**Theorem 3.2** (Exponential upper bound, see Theorem D.1). *For any $\alpha = (\alpha_0, \alpha_1, \ldots, \alpha_m) \in \mathbb{R}_{++}^{m+1}$ define $\overline{p} \in \Delta_m$ such that $\overline{p}(\ell) = \alpha_\ell/\overline{\alpha}, \ell = 0, \ldots, m$, where $\overline{\alpha} = \sum_{j=0}^m \alpha_j$. Then for any $f : \{0, \ldots, m\} \to [0, b]$ and $0 < \mu < b$, we have*

$$\mathbb{P}_{w \sim \mathcal{D}ir(\alpha)}[wf \geq \mu] \leq \exp(-\overline{\alpha}\,\mathcal{K}_{inf}(\overline{p}, \mu, f)).$$

The second result is devoted to a tight Gaussian lower bound for the distribution of a linear function of Dirichlet random vector. Here we follow the ideas of Alfers and Dinges [1984] and use the exact expression for the density of a linear form of Dirichlet random vector derived by Tiapkin et al. [2022].

**Theorem 3.3** (Gaussian lower bound, see Theorem D.2). *For any $\alpha = (\alpha_0 + 1, \alpha_1, \ldots, \alpha_m) \in \mathbb{R}_{++}^{m+1}$, define $\overline{p} \in \Delta_m$ such that $\overline{p}(\ell) = \alpha_\ell/\overline{\alpha}, \ell = 0, \ldots, m$, where $\overline{\alpha} = \sum_{j=0}^m \alpha_j$. Fix $\varepsilon \in (0, 1)$ and assume that $\alpha_0 \geq c(\varepsilon) + \log_{17/16}(\overline{\alpha})$ for $c(\varepsilon)$ defined in (11), Appendix D, and $\overline{\alpha} \geq 2\alpha_0$. Then for any $f : \{0, \ldots, m\} \to [0, b_0]$ such that $f(0) = b_0$, $f(j) \leq b < b_0/2, j \in \{1, \ldots, m\}$ and $\mu \in (\overline{p}f, b_0)$,*

$$\mathbb{P}_{w \sim \mathcal{D}ir(\alpha)}[wf \geq \mu] \geq (1 - \varepsilon)\mathbb{P}_{g \sim \mathcal{N}(0,1)}\left[g \geq \sqrt{2\overline{\alpha}\,\mathcal{K}_{inf}(\overline{p}, \mu, f)}\right].$$

We emphasize that increasing the parameter $\alpha_0$ corresponding to the largest value of $f$ by 1 is crucial. The same technique was used by Alfers and Dinges [1984] to derive a lower bound on the tails of the Beta distribution.

*Remark* 3.4. We stress that the anti-concentration inequality of Tiapkin et al. [2022, Theorem D.2] is not sufficient for our purposes; their additional factor $\overline{\alpha}^{-3/2}$ in front of the exponent makes it unusable for the analysis of OPSRL. Indeed, this inequality would imply $\widetilde{\mathcal{O}}(T^{3/2})$ samples from the inflated posterior in order to get optimism with high-probability, whereas with our refined bound (Theorem 3.3) we only need $\widetilde{\mathcal{O}}(1)$ posterior samples.

*Proof sketch of Theorem 3.3.* We start from the integral representation for the density by Tiapkin et al. [2022, Proposition D.3]. Define $Z \triangleq wf$ for $w \sim \mathcal{D}ir(\alpha_0 + 1, \alpha_1, \ldots, \alpha_m)$, then for any $u \in (0, b_0)$,

$$p_Z(u) = \frac{\overline{\alpha}}{2\pi} \int_{\mathbb{R}} (1 + \mathrm{i}(b_0 - u)s)^{-1} \prod_{j=0}^m (1 + \mathrm{i}(f(j) - u)s)^{-\alpha_j} \mathrm{d}s.$$

One additional term $(1 + \mathrm{i}(b_0 - u)s)^{-1}$ comes from increasing the parameter $\alpha_0$ by 1 corresponding to the value $f(0) = b_0$.

In the same spirit as it was done by Tiapkin et al. [2022], we apply the method of saddle point (see Fedoryuk, 1977, Olver, 1997) to the complex integral above. Informally, for $\alpha_0, \overline{\alpha}, b_0$ large enough the following approximation holds

$$p_Z(u) \approx \sqrt{\frac{\overline{\alpha}}{2\pi\sigma^2(1 - \lambda^\star(b_0 - u))^2}} \exp(-\overline{\alpha}\,\mathcal{K}_{\inf}(\overline{p}, u, f)),$$

where $\lambda^\star$ is the unique solution to the problem (2) and $\sigma^2 = \mathbb{E}_{X \sim \overline{p}}\left[\left(\frac{f(X) - u}{1 - \lambda^\star(f(X) - u)}\right)^2\right]$. The formal statement can be found in Lemma D.5 of Appendix D.

Next we perform a change of variables $t^2/2 = \mathcal{K}_{\inf}(\overline{p}, u, f)$ in the above expression to get

$$\mathbb{P}_{w \sim \mathcal{D}ir(\alpha_0 + 1, \alpha_1, \ldots, \alpha_m)}[wf \geq \mu] \approx \int_\mu^{b_0} \sqrt{\frac{\overline{\alpha}}{2\pi\sigma^2(1 - \lambda^\star(b_0 - u))^2}} \exp(-\overline{\alpha}\,\mathcal{K}_{\inf}(\overline{p}, u, f))\mathrm{d}u$$

$$\approx \int_{\sqrt{2\,\mathcal{K}_{\inf}(\overline{p}, \mu, f)}}^\infty D(u(t))\phi(t|0, \overline{\alpha})\mathrm{d}t,$$

where $\phi(x|\mu, \sigma^2)$ is a density of $\mathcal{N}(\mu, \sigma^2)$ and $D(u)$ is a weight function bounded from below by 1 (see Lemma D.6 of Appendix D). This lower bound on $D(u)$ concludes the proof. $\qquad\square$

**Comparison with anti-concentration bound by Agrawal and Jia [2017b]**  We emphasise that our technique of deriving a Gaussian-like lower bound is substantially different from the methodology used by Agrawal and Jia [2017b]. The latter one was based on reduction of a weighted sum of Dirichlet random vector to a weighted sum of independent Beta distributed random variables and a subsequent application of the Berry-Esseen inequality, whereas our approach relies on the integral representation for the density of the corresponding linear projection of Dirichlet random vector.

In particular, the Berry-Esseen inequality is likely to be very coarse since it uses only the first three moments of the distribution and therefore generates an additional $S$-factor. At the same time, our analysis is much better fitted to the Dirichlet distribution and provides a very tight lower bound. The tightness of our bounds can be checked by comparing it to a similar result for the beta distribution derived in Alfers and Dinges [1984].

**Step 2. Optimism**  Next, we apply Theorem 3.3 to prove that the estimate of Q-function $\overline{Q}_h^t$ is optimistic with high probability for our choice of inflation parameter $\kappa$ and a number of posterior samples $J$: $\overline{Q}_h^t(s, a) \geq Q_h^\star(s, a)$ for any $(s, a, h, t) \in \mathcal{S} \times \mathcal{A} \times [H] \times [T]$.

We show that the inequalities $\max_{j \in [J]}\{\widetilde{p}_h^{t,j}V_{h+1}^\star(s, a)\} \geq p_h V_{h+1}^\star(s, a)$ hold for all $(s, a, h, t) \in \mathcal{S} \times \mathcal{A} \times [H] \times [T]$ with high probability. First, we notice that $\widetilde{p}_h^{t,j}(s, a) \sim \mathcal{D}ir(\alpha_0 + 1, \alpha_1, \ldots, \alpha_S)$ for $\alpha_0 = n_0/\kappa - 1, \alpha_i = n_h^t(s_i|s, a)/\kappa$ and $\overline{\alpha} = (\overline{n}_h^t(s, a) - \kappa)/\kappa$. Additionally, define a probability distribution $q \in \Delta_S$ such that $q(i) = \alpha_i/\overline{\alpha}$. This distribution slightly differs from $\overline{p}_h^t(s, a)$ because of an additional $+1$ in the parameters of the Dirichlet distribution. Next, we may apply Theorem 3.3 with $\varepsilon = 1/2$ and a proper choice of $n_0 = n_0(\varepsilon)$,

$$\mathbb{P}_{\widetilde{p}_h^{t,j}(s,a) \sim \mathcal{D}ir(\alpha_0 + 1, \alpha_1, \ldots, \alpha_S)}\left[\widetilde{p}_h^{t,j}V_{h+1}^\star(s, a) \geq p_h V_{h+1}^\star(s, a)\right] \geq \frac{1}{2}\left(1 - \Phi\left(\sqrt{\frac{2\zeta}{\kappa}}\right)\right),$$

where $\zeta \triangleq (\overline{n}_h^t - \kappa)\,\mathcal{K}_{\inf}(q, p_h V_{h+1}^\star(s, a), V_{h+1}^\star)$ and $\Phi(\cdot)$ is a cumulative distribution function (CDF) of a standard normal distribution. By a concentration argument we have

$$\zeta \leq n_h^t\,\mathcal{K}_{\inf}(\widehat{p}_h^t(s, a), p_h V_{h+1}^\star(s, a), V_{h+1}^\star) \leq \kappa/2,$$

with high probability for an appropriate choice of $\kappa = \widetilde{\mathcal{O}}(1)$. For this step of the proof the presence of the inflation parameter $\kappa$ is crucial: this parameter increases the variance of $\widetilde{p}_h^{t,j}(s, a)$ to ensure that the above inequality holds with a constant probability. Next, by taking the maximum over $J = \mathcal{O}(\log(SATH/\delta))$ samples and applying union bound, we guarantee that the inequality $\max_{j \in [J]}\{\widetilde{p}_h^{t,j}V_{h+1}^\star(s, a)\} \geq p_h V_{h+1}^\star(s, a)$ holds simultaneously for all

$(s, a, h, t) \in \mathcal{S} \times \mathcal{A} \times [H] \times [T]$ with probability at least $1 - \delta/2$. The formal statement and the proof could be found in Proposition B.4 of Appendix B.2.

Finally, the standard backward induction over $h \in [H]$ concludes optimism. Indeed, the base of induction $h = H + 1$ is trivial. Next, by the Bellman equations for $\overline{Q}_h^t$ and $Q_h^\star$ we have

$$\overline{Q}_h^t(s, a) - Q_h^\star(s, a) = \max_{j \in [J]} \{\widetilde{p}_h^{t,j} \overline{V}_{h+1}^t(s, a)\} - p_h V_{h+1}^\star(s, a).$$

The induction hypothesis implies $\overline{V}_{h+1}^t(s') \geq \overline{Q}_{h+1}^t(s', \pi^\star(s')) \geq Q_{h+1}^\star(s', \pi^\star(s')) = V_{h+1}^\star(s')$ for any $s' \in \mathcal{S}$. Hence,

$$\overline{Q}_h^t(s, a) - Q_h^\star(s, a) \geq \max_{j \in [J]} \{\widetilde{p}_h^{t,j} V_{h+1}^\star(s, a)\} - p_h V_{h+1}^\star(s, a) \geq 0$$

with probability at least $1 - \delta/2$.

**Step 3. Regret bound** The rest of proof directly follows Azar et al. [2017], where UCBVI algorithm with Bernstein bonuses was analyzed. By the optimism, we have

$$\mathfrak{R}^T = \sum_{t=1}^T [V_1^\star(s_1) - V_1^{\pi^t}(s_1)] \leq \sum_{t=1}^T \delta_1^t,$$

where $\delta_h^t \triangleq \overline{V}_h^{t-1}(s_h^t) - V_h^{\pi^t}(s_h^t)$. The quantity $\delta_h^t$ can be decomposed as follows using the Bellman equation for $V^{\pi^t}$ and $\overline{Q}_h^{t-1}$,

$$\delta_h^t = \overline{Q}_h^{t-1}(s_h^t, a_h^t) - Q_h^{\pi^t}(s_h^t, a_h^t) = \max_{j \in [J]} \left\{ \widetilde{p}_h^{t-1,j} \overline{V}_{h+1}^{t-1}(s_h^t, a_h^t) \right\} - p_h V_{h+1}^{\pi^t}(s_h^t, a_h^t)$$

$$= \underbrace{\max_{j \in [J]} \left\{ \widetilde{p}_h^{t-1,j} \overline{V}_{h+1}^{t-1}(s_h^t, a_h^t) \right\} - \overline{p}_h^{t-1} \overline{V}_{h+1}^{t-1}(s_h^t, a_h^t)}_{(\mathbf{A})} + \underbrace{[\overline{p}_h^{t-1} - \widehat{p}_h^{t-1}] \overline{V}_{h+1}^{t-1}(s_h^t, a_h^t)}_{(\mathbf{B})}$$

$$+ \underbrace{[\widehat{p}_h^{t-1} - p_h] [\overline{V}_{h+1}^{t-1} - V_{h+1}^\star](s_h^t, a_h^t)}_{(\mathbf{C})} + \underbrace{[\widehat{p}_h^{t-1} - p_h] V_{h+1}^\star(s_h^t, a_h^t)}_{(\mathbf{D})}$$

$$+ \underbrace{p_h [\overline{V}_{h+1}^{t-1} - V_{h+1}^{\pi^t}](s_h^t, a_h^t) - [\overline{V}_{h+1}^{t-1} - V_{h+1}^{\pi^t}](s_{h+1}^t)}_{\xi_h^t} + \delta_{h+1}^t.$$

The terms $(\mathbf{C})$, $(\mathbf{D})$, and $\xi_h^t$ are standard in the analysis of the optimistic algorithms. The term $(\mathbf{B})$ could be upper-bounded by $\frac{r_0 \cdot n_0 \cdot H}{\overline{n}_h^{t-1}(s_h^t, a_h^t)}$ and turns out to be one of second-order terms. The analysis of $(\mathbf{A})$ is novel and requires application of the Bernstein inequality for Dirichlet distributions that follows from Theorem 3.2 and is spelled out in the following lemma.

**Lemma 3.5** (see Lemma C.6 in Appendix C). *For any* $\alpha = (\alpha_0, \alpha_1, \ldots, \alpha_m) \in \mathbb{R}_{++}^{m+1}$ *define* $\overline{p} \in \Delta_m$ *such that* $\overline{p}(\ell) = \alpha_\ell / \overline{\alpha}, \ell = 0, \ldots, m$, *where* $\overline{\alpha} = \sum_{j=0}^m \alpha_j$. *Then for any* $f \colon \{0, \ldots, m\} \to [0, b]$ *such that* $f(0) = b$ *and* $\delta \in (0, 1)$,

$$\mathbb{P}_{w \sim \mathcal{D}ir(\alpha)} \left[ wf \geq \overline{p}f + 2\sqrt{\frac{\mathrm{Var}_{\overline{p}}(f) \log(1/\delta)}{\overline{\alpha}}} + \frac{3b \cdot \log(1/\delta)}{\overline{\alpha}} \right] \leq \delta.$$

As opposed to Lemma C.8 of Tiapkin et al. [2022], the last result applies to Dirichlet distributions with non-integer parameters as in our case (due to the presence of the inflation parameter $\kappa$). Therefore, we see that the term $(\mathbf{A})$ can be upper bounded by a quantity which has the same role as in the analysis of UCBVI. After using the Bernstein bound, the rest of the proof follows from the analysis of UCBVI with the Bernstein bonuses and Bayes-UCBVI; see Azar et al. [2017] and Tiapkin et al. [2022].

## 4 Experiments

In this section we provide experiment to compare OPSRL with some baselines on simple tabular environment; see details in Appendix G. In particular, we illustrate that OPSRL is competitive with the original PSRL algorithm and outperforms bonus-based algorithms such as UCBVI.

**Baselines**   We compare OPSRL with the following baselines: UCBVI (with Hoeffding-type bonuses) and UCBVI-B (with Bernstein-type bonuses) Azar et al. [2017], PSRL Osband et al. [2013], and RLSVI Osband et al. [2016b]. See Appendix G for full details on parameters for OPSRL and baselines.

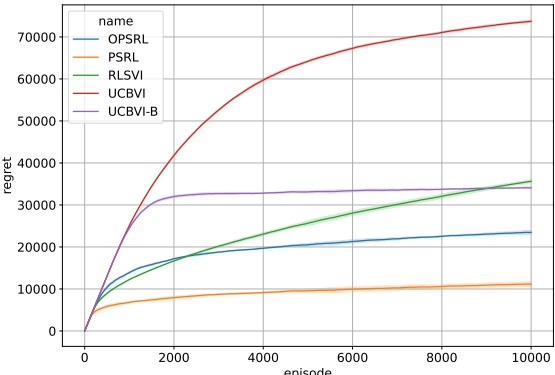

Figure 1: Regret of OPSRL and baselines on grid-world environment with 100 states and 4 action for $H = 50$ an transitions noise 0.2. We show average over 4 seeds.

**Results**   In Figure 1, we plot the regret of the various baselines and OPSRL in the grid world environment. In this experiment, we observe that OPSRL achieves competitive results with respect to PSRL. It is not completely surprising since they share the same Bayesian model on the transitions up to the prior. We shall elaborate more on the influence of the prior in Appendix G. We also note that OPSRL outperforms UCBVI and RLSVI. This difference may be explained by the fact that OPSRL's optimism implies (in the worst case) KL bonuses as in Filippi et al. [2010]. The KL bonuses are stronger than Bernstein bonuses, see Lemma E.1, because they somehow rely on all moments of the empirical distribution rather than the first two moments as in the case of Bernstein bonuses or first moments for Hoeffding bonuses or for the variance of the Gaussian noise in RLSVI. Note also that in OPSRL, we do not have to solve the complex convex program to compute the KL bonuses Filippi et al. [2010], which could be computationally intensive.

## 5   Conclusion

In this work, we presented the OPSRL algorithm which can be viewed as a simple optimistic variant of the PSRL algorithm. Notably, OPSRL only needs $\widetilde{\mathcal{O}}(1)$ posterior samples per state-action. We proved that the regret of OPSRL is upper-bounded with high probability by $\widetilde{\mathcal{O}}(\sqrt{H^3 SAT})$, matching the problem-independent lower-bound of order $\Omega(\sqrt{H^3 SAT})$ for $T$ large enough and up to terms poly-logarithmic in $H, S, A,$ and $T$. While our work addresses the open questions raised by Agrawal and Jia [2017b] in the episodic setting, obtaining the same results in the infinite-horizon average reward setting remains an open issue. We believe that it is possible to adapt our analysis to this other setting up to some technical adjustments. Ultimately, another open question, is to obtain a high-probability regret bound for PSRL, that is, when using only a *single* posterior sample and not inflating the posterior. As a further future research direction we believe it could be interesting to obtain a model-free algorithm that relies on the same mechanism as OPSRL for exploration. Indeed, such an algorithm could avoid the use of complicated bonuses adopted by the current model-free algorithms while reducing the memory complexity of OPSRL.

## **Acknowledgments and Disclosure of Funding**

D. Belomestny acknowledges the financial support from Deutsche Forschungsgemeinschaft (DFG), Grant Nr. 497300407. The work of D. Tiapkin, D. Belomestny and A. Naumov was prepared within the framework of the HSE University Basic Research Program. Pierre Ménard acknowledges the support of the Chaire SeqALO (ANR-20-CHIA-0020-01).

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
