# Appendix

## Table of Contents

# A  Notation

Table 1: Table of notation use throughout the paper

| Notation | Meaning |
|---|---|
| $\mathcal{S}$ | state space of size $S$ |
| $\mathcal{A}$ | action space of size $A$ |
| $H$ | length of one episode |
| $T$ | number of episodes |
| $J$ | number of posterior samples |
| $r_h(s,a)$ | reward |
| $p_h(s'\mid s,a)$ | probability transition |
| $Q_h^\pi(s,a)$ | Q-function of a given policy $\pi$ at step $h$ |
| $V_h^\pi(s)$ | V-function of a given policy $\pi$ at step $h$ |
| $Q_h^\star(s,a)$ | optimal Q-function at step $h$ |
| $V_h^\star(s)$ | optimal V-function at step $h$ |
| $\mathfrak{R}^T$ | regret |
| $n_0$ | number of pseudo-samples |
| $s_0$ | pseudo-state |
| $r_0$ | pseudo-reward |
| $\kappa$ | posterior inflation parameter |
| $s_h^t$ | state that was visited at $h$ step during $t$ episode |
| $a_h^t$ | action that was picked at $h$ step during $t$ episode |
| $n_h^t(s,a)$ | number of visits of state-action $n_h^t(s,a) = \sum_{k=1}^t \mathbb{1}\{(s_h^k, a_h^k) = (s,a)\}$ |
| $n_h^t(s'\mid s,a)$ | number of transition to $s'$ from state-action $n_h^t(s'\mid s,a) = \sum_{k=1}^t \mathbb{1}\{(s_h^k, a_h^k, s_{h+1}^k) = (s,a,s')\}$. |
| $\overline{n}_h^t(s,a)$ | pseudo number of visits of state-action $\overline{n}_h^t(s,a) = n_h^t(s,a) + n_0$ |
| $\overline{n}_h^t(s'\mid s,a)$ | pseudo number of transition to $s'$ from state-action $\overline{n}_h^t(s'\mid s,a) = n_h^t(s'\mid s,a) + \mathbb{1}\{s' = s_0\} \cdot n_0$ |
| $\widehat{p}_h^t(s'\mid s,a)$ | empirical probability transition $\widehat{p}_h^t(s'\mid s,a) = n_h^t(s'\mid s,a)/n_h^t(s,a)$ |
| $\overline{p}_h^t(s'\mid s,a)$ | pseudo-empirical probability transition $\overline{p}_h^t(s'\mid s,a) = \overline{n}_h^t(s'\mid s,a)/\overline{n}_h^t(s,a)$ |
| $\overline{Q}_h^t(s,a)$ | upper approximation of the optimal Q-value |
| $\overline{V}_h^t(s,a)$ | upper approximation of on the optimal V-value |
| $\mathbb{R}_+$ | non-negative real numbers |
| $\mathbb{R}_{++}$ | positive real numbers |
| $\mathbb{N}_{++}$ | positive natural numbers |
| $[n]$ | set $\{1, 2, \ldots, n\}$ |
| $\Delta_d$ | $d$-dimensional probability simplex: $\Delta_d = \{x \in \mathbb{R}_+^{d+1} : \sum_{j=0}^d x_j = 1\}$ |
| $\mathbf{1}^N$ | vector of dimension $N$ with all entries one is $\mathbf{1}^N \triangleq (1, \ldots, 1)$ |
| $\|x\|_1$ | $\ell_1$-norm of vector $\|x\|_1 = \sum_{j=1}^m |x_j|$ |
| $\|x\|_2$ | $\ell_2$-norm of vector $\|x\|_2 = \sqrt{\sum_{j=1}^m x_j^2}$ |
| $\|f\|_2$ | for $f : \mathsf{X} \to \mathbb{R}$, where $|\mathsf{X}| < \infty$ define $\|f\|_2 = \sqrt{\sum_{x \in \mathsf{X}} f^2(x)}$ |

Let $(\mathsf{X}, \mathcal{X})$ be a measurable space and $\mathcal{P}(\mathsf{X})$ be the set of all probability measures on this space. For $p \in \mathcal{P}(\mathsf{X})$ we denote by $\mathbb{E}_p$ the expectation w.r.t. $p$. For random variable $\xi : \mathsf{X} \to \mathbb{R}$ notation $\xi \sim p$ means $\mathrm{Law}(\xi) = p$. We also write $\mathbb{E}_{\xi \sim p}$ instead of $\mathbb{E}_p$. For independent (resp. i.i.d.) random variables $\xi_\ell \overset{\mathrm{ind}}{\sim} p_\ell$ (resp. $\xi_\ell \overset{\mathrm{i.i.d}}{\sim} p$), $\ell = 1, \ldots, d$, we will write $\mathbb{E}_{\xi_\ell \overset{\mathrm{ind}}{\sim} p_\ell}$ (resp. $\mathbb{E}_{\xi_\ell \overset{\mathrm{i.i.d}}{\sim} p}$), to denote expectation w.r.t. product measure on $(\mathsf{X}^d, \mathcal{X}^{\otimes d})$. For any $p, q \in \mathcal{P}(\mathsf{X})$ the Kullback-Leibler divergence $\mathrm{KL}(p, q)$ is given by

$$\mathrm{KL}(p, q) \triangleq \begin{cases} \mathbb{E}_p\left[\log \frac{\mathrm{d}p}{\mathrm{d}q}\right], & p \ll q, \\ +\infty, & \text{otherwise.} \end{cases}$$

For any $p \in \mathcal{P}(\mathsf{X})$ and $f : \mathsf{X} \to \mathbb{R}$, $pf = \mathbb{E}_p[f]$. In particular, for any $p \in \Delta_d$ and $f : \{0, \ldots, d\} \to \mathbb{R}$, $pf = \sum_{\ell=0}^d f(\ell)p(\ell)$. Define $\mathrm{Var}_p(f) = \mathbb{E}_{s' \sim p}\left[(f(s') - pf)^2\right] = p[f^2] - (pf)^2$. For any $(s,a) \in \mathcal{S}$, transition kernel $p(s,a) \in \mathcal{P}(\mathcal{S})$ and $f : \mathcal{S} \to \mathbb{R}$ define $pf(s,a) = \mathbb{E}_{p(s,a)}[f]$ and $\mathrm{Var}_p[f](s,a) = \mathrm{Var}_{p(s,a)}[f]$.

We write $f(S, A, H, T) = \mathcal{O}(g(S, A, H, T, \delta))$ if there exist $S_0, A_0, H_0, T_0, \delta_0$ and constant $C_{f,g}$ such that for any $S \geq S_0, A \geq A_0, H \geq H_0, T \geq T_0, \delta < \delta_0, f(S, A, H, T, \delta) \leq C_{f,g} \cdot g(S, A, H, T, \delta)$. We write $f(S, A, H, T, \delta) = \widetilde{\mathcal{O}}(g(S, A, H, T, \delta))$ if $C_{f,g}$ in the previous definition is poly-logarithmic in $S, A, H, T, 1/\delta$.

For $\lambda > 0$, we define $\mathcal{E}(\lambda)$ as an exponential distribution with a parameter $\lambda$. For $k, \theta > 0$ define $\Gamma(k, \theta)$ as a gamma-distribution with a shape parameter $k$ and a rate parameter $\theta$. For set $\mathsf{X}$ such that $|\mathsf{X}| < \infty$ define $\mathcal{U}\mathrm{nif}(\mathsf{X})$ as a uniform distribution over this set. In particular, $\mathcal{U}\mathrm{nif}[N]$ is a uniform distribution over a set $[N]$.

We fix a function $f : \{1, \dots, m\} \mapsto [0, b]$ and recall the definition of the minimum Kullback-Leibler divergence for $p \in \Delta_{m-1}$ and $u \in \mathbb{R}$

$$\mathcal{K}_{\mathrm{inf}}(p, u, f) \triangleq \inf\{\mathrm{KL}(p, q) : q \in \Delta_{m-1}, qf \geq u\}.$$

As the Kullback-Leibler divergence this quantity admits a variational formula by Lemma 18 of Garivier et al. [2018] up to rescaling for any $u \in (0, b)$

$$\mathcal{K}_{\mathrm{inf}}(p, u, f) = \max_{\lambda \in [0, 1/(b-u)]} \mathbb{E}_{X \sim p}[\log(1 - \lambda(f(X) - u))].$$

## B  Proof of regret bound for OPSRL

### B.1  Concentration events

Let $\beta^\star, \beta^{\mathrm{KL}}, \beta^{\mathrm{conc}}, \beta^{\mathrm{Var}}\colon (0,1)\times\mathbb{N}\to\mathbb{R}_+$, $\beta^{\mathcal{D}\mathrm{ir}}\colon (0,1)\times\mathbb{N}\times\mathbb{N}\to\mathbb{R}_+$, and $\beta\colon(0,1)\to\mathbb{R}_+$ be some function defined later in Lemma B.1. We define the following favorable events,

$$\mathcal{E}^\star(\delta) \triangleq \left\{ \forall t\in\mathbb{N}, \forall h\in[H], \forall(s,a)\in\mathcal{S}\times\mathcal{A}: \right.$$

$$\left. \mathcal{K}_{\inf}(\widehat{p}_h^t(s,a), p_h V_{h+1}^\star(s,a), V_{h+1}^\star) \leq \frac{\beta^\star(\delta, n_h^t(s,a))}{n_h^t(s,a)} \right\},$$

$$\mathcal{E}^{\mathrm{KL}}(\delta) \triangleq \left\{ \forall t\in\mathbb{N}, \forall h\in[H], \forall(s,a)\in\mathcal{S}\times\mathcal{A}: \right.$$

$$\left. \mathrm{KL}(\widehat{p}_h^t(s,a), p_h(s,a)) \leq \frac{S\cdot\beta^{\mathrm{KL}}(\delta, n_h^t(s,a))}{n_h^t(s,a)} \right\},$$

$$\mathcal{E}^{\mathrm{conc}}(\delta) \triangleq \left\{ \forall t\in\mathbb{N}, \forall h\in[H], \forall(s,a)\in\mathcal{S}\times\mathcal{A}: \right.$$

$$\left. |(\widehat{p}_h^t - p_h)V_{h+1}^\star(s,a)| \leq \sqrt{2\mathrm{Var}_{p_h}(V_{h+1}^\star)(s,a)\frac{\beta(\delta, n_h^t(s,a))}{n_h^t(s,a)}} + 3H\frac{\beta(\delta, n_h^t(s,a))}{n_h^t(s,a)} \right\},$$

$$\mathcal{E}^{\mathcal{D}\mathrm{ir}}(\delta) \triangleq \left\{ \forall t\in[T], \forall h\in[H], \forall(s,a)\in\mathcal{S}\times\mathcal{A}, \forall j\in[J]: \right.$$

$$\left. [\widetilde{p}_h^{t,j} - \overline{p}_h^t]\overline{V}_{h+1}^t(s,a) \leq 2\sqrt{\mathrm{Var}_{\overline{p}_h^t}[\overline{V}_{h+1}^t](s,a)\frac{\beta^{\mathcal{D}\mathrm{ir}}(\delta, T, J)\cdot\kappa}{\overline{n}_h^t(s,a)}} + 3r_0 H\frac{\beta^{\mathcal{D}\mathrm{ir}}(\delta, T, J)\cdot\kappa}{\overline{n}_h^t(s,a)} \right\},$$

$$\mathcal{E}^{\mathrm{Var}}(\delta) \triangleq \left\{ \forall t\in\mathbb{N}: \quad \sum_{\ell=1}^{t}\sum_{h=1}^{H}\mathrm{Var}_{p_h}[V_{h+1}^{\pi_\ell}(s_h^\ell, a_h^\ell)] \leq H^2 t + \sqrt{2H^5 t\beta^{\mathrm{Var}}(\delta, t)} + 3H^3\beta^{\mathrm{Var}}(\delta, t) \right\},$$

$$\mathcal{E}(\delta) \triangleq \left\{ \sum_{t=1}^{T}\sum_{h=1}^{H}\left|p_h[\overline{V}_{h+1}^{t-1} - V_{h+1}^{\pi^t}](s_h^t, a_h^t) - [\overline{V}_{h+1}^{t-1} - V_{h+1}^{\pi^t}](s_{h+1}^t)\right| \leq 2r_0 H\sqrt{2HT\beta(\delta)}, \right.$$

$$\sum_{t=1}^{T}\sum_{h=1}^{H}(1-1/H)^{H-h+1}\left|p_h[\overline{V}_{h+1}^{t-1} - V_{h+1}^{\pi^t}](s_h^t, a_h^t)\right.$$

$$\left.\left. - [\overline{V}_{h+1}^{t-1} - V_{h+1}^{\pi^t}](s_{h+1}^t)\right| \leq 2er_0 H\sqrt{2HT\beta(\delta)}, \right\}.$$

We also introduce the intersection of these events, $\mathcal{G}^{\mathrm{conc}}(\delta) \triangleq \mathcal{E}^\star(\delta)\cap\mathcal{E}^{\mathrm{KL}}(\delta)\cap\mathcal{E}^{\mathrm{conc}}(\delta)\cap\mathcal{E}^{\mathcal{D}\mathrm{ir}}(\delta)\cap\mathcal{E}^{\mathrm{Var}}(\delta)\cap\mathcal{E}(\delta)$. We prove that for the right choice of the functions $\beta^\star, \beta^{\mathrm{KL}}, \beta^{\mathrm{conc}}, \beta, \beta^{\mathrm{Var}}$, the above events hold with high probability.

**Lemma B.1.** *For any $\delta\in(0,1)$ and for the following choices of functions $\beta$,*

$$\beta^\star(\delta, n) \triangleq \log(12SAH/\delta) + 3\log(e\pi(2n+1)),$$

$$\beta^{\mathrm{KL}}(\delta, n) \triangleq \log(12SAH/\delta) + \log(e(1+n)),$$

$$\beta^{\mathrm{conc}}(\delta, n) \triangleq \log(12SAH/\delta) + \log(4e(2n+1)),$$

$$\beta^{\mathcal{D}\mathrm{ir}}(\delta, t, J) \triangleq \log(12SAHt/\delta) + \log(J),$$

$$\beta^{\mathrm{Var}}(\delta, t) \triangleq \log(48e(2t+1)/\delta),$$

$$\beta(\delta) \triangleq \log(48/\delta),$$

*it holds that*

$$\mathbb{P}[\mathcal{E}^\star(\delta)] \geq 1 - \delta/12, \qquad \mathbb{P}[\mathcal{E}^{\mathrm{KL}}(\delta)] \geq 1 - \delta/12, \qquad \mathbb{P}[\mathcal{E}^{\mathrm{conc}}(\delta)] \geq 1 - \delta/12,$$

$$\mathbb{P}[\mathcal{E}^{\mathcal{D}\mathrm{ir}}(\delta)] \geq 1 - \delta/12, \qquad \mathbb{P}[\mathcal{E}^{\mathrm{Var}}(\delta)] \geq 1 - \delta/12, \qquad \mathbb{P}[\mathcal{E}(\delta)] \geq 1 - \delta/12.$$

*In particular,* $\mathbb{P}[\mathcal{G}^{\mathrm{conc}}(\delta)] \geq 1 - \delta/2.$

*Remark* B.2. Since we take $J \triangleq \Theta(\log(SAHT/\delta))$, all functions $\beta$ are logarithmic in $S, A, H, T, \delta$.

*Proof.* $\mathbb{P}[\mathcal{E}^\star(\delta)] \geq 1 - \delta/12$ follows from Theorem C.4. Applying Theorem C.1 and the union bound over $h \in [H], (s,a) \in \mathcal{S} \times \mathcal{A}$ we get $\mathbb{P}[\mathcal{E}^{\mathrm{KL}}(\delta)] \geq 1 - \delta/12$. Next, by Lemma C.6 and the union bound over $h \in [H], t \in [T], (s,a) \in \mathcal{S} \times \mathcal{A}, j \in [J]$ we conclude $\mathbb{P}[\mathcal{E}^{\mathcal{D}\mathrm{ir}}(\delta)] \geq 1 - \delta/12$. Theorem C.5 and the union bound over $h \in [H], (s,a) \in \mathcal{S} \times \mathcal{A}$ yield $\mathbb{P}[\mathcal{E}^{\mathrm{conc}}(\delta)] \geq 1 - \delta/12$. By Lemma B.2 by Tiapkin et al. [2022] we have $\mathbb{P}[\mathcal{E}^{\mathrm{Var}}(\delta)] \geq 1 - \delta/12$.

To estimate $\mathbb{P}[\mathcal{E}(\delta)]$ one may apply Azuma-Hoeffding inequality. Define the following sequences for all $t \in [T], h \in [H]$

$$\bar{Z}_{t,h} \triangleq \overline{V}_{h+1}^{t-1}(s_{h+1}^t) - V_{h+1}^*(s_{h+1}^t) - p_h[\overline{V}_{h+1}^{t-1} - V_{t+1}^*](s_h^t, a_h^t),$$

$$\tilde{Z}_{t,h} \triangleq (1 - 1/H)^{H-h+1}\left(\overline{V}_{h+1}^{t-1}(s_{h+1}^t) - V_{h+1}^*(s_{h+1}^t) - p_h[\overline{V}_{h+1}^{t-1} - V_{h+1}^*](s_h^t, a_h^t)\right).$$

It is easy to see that these sequences form a martingale-difference w.r.t filtration $\mathcal{F}_{t,h} = \sigma\{\{(s_{h'}^\ell, a_{h'}^\ell), \ell < t, h' \in [H]\} \cup \{(s_{h'}^t, a_{h'}^t), h' \leq h\}\}$. Moreover, $|\bar{Z}_{t,h}| \leq 2r_0 H, |\tilde{Z}_{t,h}| \leq 2er_0 H$ for all $t \in [T]$ and $h \in [H]$. Hence, the Azuma-Hoeffding inequality implies

$$\mathbb{P}\left(\left|\sum_{t=1}^T \sum_{h=1}^H \bar{Z}_{t,h}\right| > 2r_0 H\sqrt{2tH \cdot \beta(\delta)}\right) \leq 2\exp(-\beta(\delta)) = \delta/24,$$

$$\mathbb{P}\left(\left|\sum_{t=1}^T \sum_{h=1}^H \bar{Z}_{t,h}\right| > 2er_0 H\sqrt{2tH \cdot \beta(\delta)}\right) \leq 2\exp(-\beta(\delta)) = \delta/24.$$

By the union bound, $\mathbb{P}[\mathcal{E}(\delta)] \geq 1 - \delta/12$. $\qquad\square$

Next we reproduce proof of important corollary of Lemma B.1.

**Lemma B.3.** *Assume conditions of Lemma B.1. Then on the event* $\mathcal{E}^{\mathrm{KL}}(\delta)$, *for any* $f: \mathcal{S} \to [0, r_0 H]$, $t \in \mathbb{N}, h \in [H], (s,a) \in \mathcal{S} \times \mathcal{A}$,

$$(\widehat{p}_h^t - p_h)f(s,a) \leq \frac{1}{H}p_h f(s,a) + \frac{5r_0 H^2 S \cdot \beta^{\mathrm{KL}}(\delta, n_h^t(s,a))}{n_h^t(s,a)},$$

$$\|\widehat{p}_h^t(s,a) - p_h(s,a)\|_1 \leq \sqrt{\frac{2S \cdot \beta^{\mathrm{KL}}(\delta, n_h^t(s,a))}{n_h^t(s,a)}}.$$

*Proof.* By application of Lemma E.1 and Lemma E.2

$$(\widehat{p}_h^t - p_h)f(s,a) \leq \sqrt{2\mathrm{Var}_{\widehat{p}_h^t}[f](s,a) \cdot \mathrm{KL}(\widehat{p}_h^t, p_h)} + \frac{2Hr_0}{3}\mathrm{KL}(\widehat{p}_h^t, p_h)$$

$$\leq 2\sqrt{\mathrm{Var}_{p_h}[f](s,a) \cdot \mathrm{KL}(\widehat{p}_h^t, p_h)} + \left(2\sqrt{2} + \frac{2}{3}\right)Hr_0 \mathrm{KL}(\widehat{p}_h^t, p_h).$$

Since $0 \leq f(s) \leq r_0 H$

$$\mathrm{Var}_{p_h}[f](s,a) \leq p_h[f^2](s,a) \leq r_0 H \cdot p_h f(s,a).$$

Finally, by a simple bound $2\sqrt{ab} \leq a + b, a, b \geq 0$, we obtain the following

$$(\widehat{p}_h^t - p_h)f(s,a) \leq \frac{1}{H}p_h f(s,a) + (H^2 + 2\sqrt{2}r_0 H + 2r_0 H/3)\mathrm{KL}(\widehat{p}_h^t, p_h)$$

$$\leq \frac{1}{H}p_h f(s,a) + 5r_0 H^2 \mathrm{KL}(\widehat{p}_h^t, p_h).$$

Definition of $\mathcal{E}^{\mathrm{KL}}(\delta)$ implies the first statement. The second statement follows directly from the combination of Pinsker's inequality and definition of $\mathcal{E}^{\mathrm{KL}}(\delta)$. $\qquad\square$

## B.2 Optimism

In this section we prove that our estimate of $Q$-function $\overline{Q}_h^t(s,a)$ is optimistic that is the event

$$\mathcal{E}_{\text{opt}} \triangleq \left\{ \forall t \in [T], h \in [H], (s,a) \in \mathcal{S} \times \mathcal{A} : \overline{Q}_h^t(s,a) \geq Q_h^\star(s,a) \right\}. \tag{3}$$

holds with high probability on the event $\mathcal{E}^\star(\delta)$.

Define constants

$$c_0 \triangleq \left( \frac{4}{\sqrt{\log(17/16)}} + 8 + \frac{49 \cdot 4\sqrt{6}}{9} \right)^2 \frac{8}{\pi} + \log_{17/16}\left( \frac{20}{32} \right) + 1, \tag{4}$$

and

$$c_J \triangleq \frac{1}{\log\left( \frac{2}{1+\Phi(1)} \right)}, \tag{5}$$

where $\Phi(\cdot)$ is a cdf of a normal distribution.

**Proposition B.4.** *Assume that* $J = \lceil c_J \cdot \log(2SAHT/\delta) \rceil$, $\kappa = 2\beta^\star(\delta, T)$, $r_0 = 2$, *and* $n_0 = \lceil (c_0 + \log_{17/16}(T/\kappa)) \cdot \kappa \rceil$. *Then on event* $\mathcal{E}^\star$ *the following event*

$$\mathcal{E}^{\text{anticonc}}(\delta) \triangleq \left\{ \forall t \in [T] \, \forall h \in [H] \, \forall (s,a) \in \mathcal{S} \times \mathcal{A} : \max_{j \in [J]}\left\{ \widetilde{p}_h^{t,j} V_{h+1}^\star(s,a) \right\} \geq p_h V_{h+1}^\star(s,a) \right\}$$

*holds with probability at least* $1 - \delta/2$.

*Proof.* First, we notice that $\widetilde{p}_h^{t,j}(s,a)$ for all fixed $t, j, h, s, a$ have a Dirichlet distribution with parameter $(\{\overline{n}_h^t(s'|s,a)/\kappa\}_{s' \in \mathcal{S}'})$ for an extended state-space $\mathcal{S}' = \{s_0\} \cup \mathcal{S}$. Therefore, we may apply Theorem D.2 with fixed $\varepsilon = 1/2$ for $f = V_{h+1}^\star$ if we have $b_0 = r_0(H-h) \geq 2(H-h) = 2b$ and

$$\frac{n_0}{\kappa} = \alpha_0 + 1 \geq c_0 + \log_{17/16}\left( \overline{n}_h^t(s,a)/\kappa \right)$$

for a constant $c_0$ defined in (4). Let us define $\alpha_0 = n_0/\kappa - 1$ and $\alpha_i = n_h^t(s_i|s,a)/\kappa$ for some ordering $s_i \in \mathcal{S}$. Then we have $\overline{\alpha} = \overline{n}_h^t(s,a)/\kappa - 1$ and $\widetilde{p}_h^{t,j} \sim \mathcal{D}\text{ir}(\alpha_0 + 1, \alpha_1, \ldots, \alpha_S)$. Define a distribution $q \in \Delta_S : q(i) = \alpha_i/\overline{\alpha}$. Then if $\overline{\alpha} \geq 2\alpha_0$ Theorem D.2 yields for any $u \geq qV_{h+1}^\star$

$$\mathbb{P}\left( \widetilde{p}_h^{t,j} V_{h+1}^\star(s,a) \geq u \right) \geq \frac{1}{2}\left( 1 - \Phi\left( \sqrt{\frac{2(\overline{n}_h^t(s,a) - \kappa)\mathcal{K}_{\inf}(q, u, V_{h+1}^\star)}{\kappa}} \right) \right), \tag{6}$$

where $\Phi$ is a cdf of a normal distribution.

Notice that if we have $u < qV_{h+1}^\star$ then the following bound also holds

$$\mathbb{P}\left( \widetilde{p}_h^{t,j} V_{h+1}^\star(s,a) \geq u \right) \geq \mathbb{P}\left( \widetilde{p}_h^{t,j} V_{h+1}^\star(s,a) \geq \overline{p}_h^t V_{h+1}^\star(s,a) \right) \geq \frac{1}{2}(1 - \Phi(0)). \tag{7}$$

Since for all $u \leq qV_{h+1}^\star$ we also have $\mathcal{K}_{\inf}\left( q, u, V_{h+1}^\star \right) = 0$, therefore (6) holds for all $u \geq 0$ and $\overline{\alpha} \geq 2\alpha_0$.

Next we need to handle the case $\overline{\alpha} < 2\alpha_0$. In this case we have $qV_{h+1}^\star > H - h$, thus for any $0 \leq u \leq H - h$

$$\mathbb{P}\left( \widetilde{p}_h^{t,j} V_{h+1}^\star(s,a) \geq u \right) \geq \mathbb{P}_{\xi \sim B(\alpha_0+1, \overline{\alpha}-\alpha_0)}(r_0(H-h)\xi \geq u) \geq \mathbb{P}_{\xi \sim B(\alpha_0+1, \overline{\alpha}-\alpha_0)}\left( \xi \geq \frac{1}{2} \right),$$

where we first apply a lower bound $V_{h+1}^\star(s) \geq 0$ for all $s \in \mathcal{S}$ and $V_{h+1}^\star(s_0) = r_0(H-h)$, and second apply a bound $u \leq H - h$. Here we may apply the result of Alfers and Dinges [1984, Theorem 1.2"] and obtain the following lower bound that is equivalent to (7)

$$\mathbb{P}\left( \widetilde{p}_h^{t,j} V_{h+1}^\star(s,a) \geq u \right) \geq \Phi\left( -\text{sign}(\alpha_0/\overline{\alpha} - 1/2) \cdot \sqrt{2\overline{\alpha}\,\text{kl}(\alpha_0/\overline{\alpha}, 1/2)} \right) \geq \frac{1}{2}(1 - \Phi(0)),$$

where we used $\alpha_0/\overline{\alpha} > 1/2$.

Thus, we may apply equation (6) for $u = p_h V_{h+1}^\star(s,a) \le H - h$ and any $\overline{\alpha}$

$$\mathbb{P}\Big(\widetilde{p}_h^{t,j} V_{h+1}^\star(s,a) \ge p_h V_{h+1}^\star(s,a)\Big) \ge \frac{1}{2}\left(1 - \Phi\left(\sqrt{\frac{2(\overline{n}_h^t(s,a) - \kappa)\,\mathcal{K}_{\inf}(q, p_h V_{h+1}^\star(s,a), V_{h+1}^\star)}{\kappa}}\right)\right).$$

By the following relation that follows from variational formula for $\mathcal{K}_{\inf}$ with rescaling of $\lambda$ to $[0,1]$

$$(\overline{n}_h^t(s,a) - \kappa)\,\mathcal{K}_{\inf}(q, u, V_{h+1}^\star) = (\overline{n}_h^t(s,a) - \kappa) \max_{\lambda \in [0,1]} \mathbb{E}_{s' \sim q}\left[\log\left(1 - \lambda \frac{V_{h+1}^\star(s') - u}{r_0(H - h) - u}\right)\right]$$

$$\le \max_{\lambda \in [0,1]}(n_0 - \kappa)\log(1 - \lambda) + (\overline{n}_h^t(s,a) - n_0) \max_{\lambda \in [0,1]} \mathbb{E}_{s' \sim \widehat{p}_h^t(s,a)}\left[\log\left(1 - \lambda \frac{V_{h+1}^\star(s') - u}{r_0(H - h) - u}\right)\right]$$

$$\le (\overline{n}_h^t(s,a) - n_0) \max_{\lambda \in [0,1]} \mathbb{E}_{s' \sim \widehat{p}_h^t(s,a)}\left[\log\left(1 - \lambda \frac{V_{h+1}^\star(s') - u}{H - h - u}\right)\right]$$

$$= (\overline{n}_h^t(s,a) - n_0)\,\mathcal{K}_{\inf}(\widehat{p}_h^t(s,a), u, V_{h+1}^\star) = n_h^t(s,a)\,\mathcal{K}_{\inf}(\widehat{p}_h^t(s,a), u, V_{h+1}^\star).$$

Thus, on the event $\mathcal{E}^\star$

$$(\overline{n}_h^t(s,a) - \kappa)\,\mathcal{K}_{\inf}\big(q, p_h V_{h+1}^\star(s,a), V_{h+1}^\star\big) \le \beta^\star(\delta, n_h^t(s,a)) \le \beta^\star(\delta, T),$$

and, as a corollary

$$\mathbb{P}\Big(\widetilde{p}_h^{t,j} V_{h+1}^\star(s,a) \ge p_h V_{h+1}^\star(s,a) \mid \mathcal{E}^\star(\delta)\Big) \ge \frac{1}{2}\left(1 - \Phi\left(\sqrt{\frac{2\beta^\star(\delta,T)}{\kappa}}\right)\right).$$

By taking $\kappa = 2\beta^\star(\delta, T)$ we have a constant probability of being optimistic

$$\mathbb{P}\Big(\widetilde{p}_h^{t,j} V_{h+1}^\star(s,a) \ge p_h V_{h+1}^\star(s,a) \mid \mathcal{E}^\star(\delta)\Big) \ge \frac{1 - \Phi(1)}{2} \triangleq \gamma.$$

Next, using a choice $J = \lceil \log(2SAHT/\delta)/\log(1/(1-\gamma)) \rceil = \lceil c_J \cdot \log(2SAHT/\delta) \rceil$

$$\mathbb{P}\left(\max_{j \in [J]}\Big\{\widetilde{p}_h^{t,j} V_{h+1}^\star(s,a)\Big\} \ge p_h V_{h+1}^\star(s,a) \mid \mathcal{E}^\star(\delta)\right) \ge 1 - (1 - \gamma)^J \ge 1 - \frac{\delta}{2SAHT}.$$

By a union bound we conclude the statement. $\qquad\square$

Next we provide a connection between $\mathcal{E}^{\text{anticonc}}(\delta)$ and $\mathcal{E}^{\text{opt}}$.

**Proposition B.5.** *For any $\delta \in (0,1)$ it holds $\mathcal{E}^{\text{opt}} \subseteq \mathcal{E}^{\text{anticonc}}(\delta)$.*

*Proof.* We proceed by a backward induction over $h$. Base of induction $h = H + 1$ is trivial. Next by Bellman equations for $\overline{Q}_h^t$ and $Q_h^\star$

$$[\overline{Q}_h^t - Q_h^\star](s,a) = \max_{j \in [J]}\Big\{\widetilde{p}_h^{t,j} \overline{V}_{h+1}^t(s,a)\Big\} - p_h V_{h+1}^\star(s,a).$$

By induction hypothesis we have $\overline{V}_{h+1}^t(s') \ge \overline{Q}_{h+1}^t(s', \pi^\star(s')) \ge Q_{h+1}^\star(s', \pi^\star(s')) = V_{h+1}^\star(s')$, thus

$$[\overline{Q}_h^t - Q_h^\star](s,a) \ge \max_{j \in [J]}\Big\{\widetilde{p}_h^{t,j} V_{h+1}^\star(s,a)\Big\} - p_h V_{h+1}^\star(s,a).$$

By the definition of event $\mathcal{E}^{\text{anticonc}}(\delta)$ we conclude the statement. $\qquad\square$

**Proposition B.6** (Optimism). *Assume that $J = \lceil c_J \cdot \log(2SAHT/\delta) \rceil$, $\kappa = 2\beta^\star(\delta, T)$, $r_0 = 2$ and $n_0 = \lceil (c_0 + \log_{17/16}(T/\kappa)) \cdot \kappa \rceil$, where $c_0$ is defined in (4) and $c_J$ is defined in (5). Then $\mathbb{P}(\mathcal{E}^{\text{opt}} \mid \mathcal{E}^\star(\delta)) \ge 1 - \delta/2$.*

## B.3 Proof of Theorem 3.1

First, we define an event $\mathcal{G}(\delta) = \mathcal{G}^{\mathrm{conc}}(\delta) \cap \mathcal{E}^{\mathrm{opt}}$ where $\mathcal{G}^{\mathrm{conc}}$ defined in Lemma B.1, and $\mathcal{E}^{\mathrm{opt}}$ defined in (3). This event is handle all required concentration and anti-concentration bounds for the proof of the regret bound. Lemma B.1 and Proposition B.6 yield the following

**Corollary B.7.** *Let conditions of Lemma B.1 and Proposition B.6 hold. Then $\mathbb{P}(\mathcal{G}(\delta)) \geq 1 - \delta$.*

Next, denote $\delta_h^t \triangleq \overline{V}_h^{t-1}(s_h^t) - V_h^{\pi^t}(s_h^t)$ and surrogate regret $\overline{\mathfrak{R}}_h^t \triangleq \sum_{t=1}^T \delta_h^t$. To simplify notations denote $N_h^t = n_h^{t-1}(s_h^t, a_h^t)$, $N_h^t(s) = n_h^{t-1}(s|s_h^t, a_h^t)$, $\overline{N}_h^t = \overline{n}_h^{t-1}(s_h^t, a_h^t)$, $\overline{N}_h^t(s) = \overline{n}_h^{t-1}(s|s_h^t, a_h^t)$. Let

$$L = \max\{n_0/\kappa, \log(TH), \beta^\star(\delta, T), \beta^{\mathrm{KL}}(\delta, T), \beta^{\mathcal{D}\mathrm{ir}}(\delta, T, J), \beta^{\mathrm{conc}}(\delta, T), \beta(\delta), \beta^{\mathrm{Var}}(\delta, T)\}. \tag{8}$$

Under conditions Lemma B.1 and Proposition B.6, $L = \mathcal{O}(\log(SATH/\delta)) = \widetilde{\mathcal{O}}(1)$, $n_0 \leq 2L^2 = \mathcal{O}(\log^2(SATH/\delta))$, and $\kappa \leq 2L$. In what follows we will follow ideas of UCBVI with the Bernstein bonuses, see Azar et al. [2017], and Bayes-UCBVI, see Tiapkin et al. [2022].

**Lemma B.8.** *Assume conditions of Theorem 3.1. Then it holds on the event $\mathcal{G}(\delta)$, for any $h \in [H]$,*

$$\overline{\mathfrak{R}}_h^T \leq U_h^T \triangleq A_h^T + B_h^T + C_h^T + 4\mathrm{e}H\sqrt{2HTL} + 2\mathrm{e}H^2 SA,$$

*where*

$$A_h^T = 3\mathrm{e}L \sum_{t=1}^T \sum_{h'=h}^H \sqrt{\mathrm{Var}_{\overline{p}_{h'}^{t-1}}[\overline{V}_{h+1}^{t-1}](s_{h'}^t, a_{h'}^t) \cdot \frac{\mathbb{1}\{N_{h'}^t > 0\}}{N_{h'}^t}},$$

$$B_h^T = \mathrm{e}\sqrt{2L} \sum_{t=1}^T \sum_{h'=h}^H \sqrt{\mathrm{Var}_{p_{h'}}[V_{h+1}^\star](s_{h'}^t, a_{h'}^t) \frac{\mathbb{1}\{N_{h'}^t > 0\}}{N_{h'}^t}},$$

$$C_h^T = 26H^2 SL^2 \sum_{t=1}^T \sum_{h'=h}^H \frac{\mathbb{1}\{N_{h'}^t > 0\}}{N_{h'}^t},$$

*and $L$ is defined in (8).*

*Proof.* Since our actions are greedy with respect to $\overline{Q}_h^{t-1}$, we have $\overline{V}_h^{t-1}(s_h^t) = \overline{Q}_h^{t-1}(s_h^t, a_h^t)$. Then by Bellman equations for $V^{\pi^t}$ and $\overline{Q}_h^{t-1}$

$$\delta_h^t = \overline{Q}_h^{t-1}(s_h^t, a_h^t) - Q_h^{\pi^t}(s_h^t, a_h^t) = \max_{j \in [J]}\{\widetilde{p}_h^{t-1,j}\overline{V}_{h+1}^{t-1}(s_h^t, a_h^t)\} - p_h V_{h+1}^{\pi^t}(s_h^t, a_h^t)$$

$$= \underbrace{\max_{j \in [J]}\{\widetilde{p}_h^{t-1,j}\overline{V}_{h+1}^{t-1}(s_h^t, a_h^t)\} - \overline{p}_h^{t-1}\overline{V}_{h+1}^{t-1}(s_h^t, a_h^t)}_{(\mathbf{A})} + \underbrace{[\overline{p}_h^{t-1} - \widehat{p}_h^{t-1}]\overline{V}_{h+1}^{t-1}(s_h^t, a_h^t)}_{(\mathbf{B})}$$

$$+ \underbrace{[\widehat{p}_h^{t-1} - p_h][\overline{V}_{h+1}^{t-1} - V_{h+1}^\star](s_h^t, a_h^t)}_{(\mathbf{C})} + \underbrace{[\widehat{p}_h^{t-1} - p_h]V_{h+1}^\star(s_h^t, a_h^t)}_{(\mathbf{D})}$$

$$+ \underbrace{p_h[\overline{V}_{h+1}^{t-1} - V_{h+1}^{\pi^t}](s_h^t, a_h^t) - [\overline{V}_{h+1}^{t-1} - V_{h+1}^{\pi^t}](s_{h+1}^t)}_{\xi_h^t} + \delta_{h+1}^t.$$

This decomposition is almost equivalent to the decomposition in the proof of Bayes-UCBVI, the main difference is $(\mathbf{A})$ and an another value of $n_0$. We notice that the term $\xi_h^t$ is an exactly the term that appears in the definition of the event $\mathcal{E}(\delta) \subseteq \mathcal{G}^{\mathrm{conc}}(\delta)$ in Lemma B.1.

Let us analyze each term in this representation under assumption $N_h^t > 0$.

**Term (A).** To handle this term, we use the event $\mathcal{E}^{\mathcal{D}\mathrm{ir}}(\delta) \subseteq \mathcal{G}^{\mathrm{conc}}(\delta)$

$$\max_{j \in [J]}\left\{\widetilde{p}_h^{t-1,j}\overline{V}_{h+1}^{t-1}(s_h^t, a_h^t)\right\} - \overline{p}_h^{t-1}\overline{V}_{h+1}^{t-1}(s_h^t, a_h^t) \le 2\sqrt{\mathrm{Var}_{\overline{p}_h^{t-1}}[\overline{V}_{h+1}^{t-1}](s_h^t, a_h^t)\frac{2L^2}{\overline{N}_h^t}} + 3r_0 H\frac{2L^2}{\overline{N}_h^t}$$

$$\le 3L\sqrt{\frac{\mathrm{Var}_{\overline{p}_h^{t-1}}[\overline{V}_{h+1}^{t-1}](s_h^t, a_h^t)}{\overline{N}_h^t}} + \frac{12HL^2}{\overline{N}_h^t}.$$

**Term (B).** To bound (B) we use directly a definition of $\overline{p}_h^{t-1}$ and $\widehat{p}_h^{t-1}$

$$[\overline{p}_h^{t-1} - \widehat{p}_h^{t-1}]\overline{V}_{h+1}^{t-1}(s_h^t, a_h^t) \le \frac{n_0 r_0 H}{\overline{N}_h^t} \le \frac{4HL^2}{\overline{N}_h^t}.$$

**Term (C).** Note that by Corollary B.7 the event $\mathcal{E}^{\mathrm{opt}}$ holds. We see that $[\overline{V}_{h+1}^{t-1} - V_{h+1}^\star]$ is a non-negative function and therefore Lemma B.3 is applicable for $f(s') = [\overline{V}_{h+1}^{t-1} - V_{h+1}^\star](s')$

$$[\widehat{p}_h^{t-1} - p_h][\overline{V}_{h+1}^{t-1} - V_{h+1}^\star](s_h^t, a_h^t) \le \frac{1}{H}p_h[\overline{V}_{h+1}^{t-1} - V_{h+1}^\star](s_h^t, a_h^t) + \frac{5r_0 H^2 S \cdot \beta^{\mathrm{KL}}(\delta, N_h^t)}{N_h^t}$$

$$\le \frac{1}{H}(\xi_h^t + \delta_h^t) + \frac{10H^2 S \cdot L}{N_h^t}.$$

**Term (D).** The bound on this term is guaranteed by the event $\mathcal{E}^{\mathrm{conc}}(\delta) \subseteq \mathcal{G}^{\mathrm{conc}}(\delta)$

$$(\widehat{p}_h^{t-1} - p_h)V_{h+1}^\star(s_h^t, a_h^t) \le \sqrt{2\mathrm{Var}_{p_h}[V_{h+1}^\star](s_h^t, a_h^t)\frac{L}{N_h^t}} + \frac{3HL}{N_h^t}.$$

All bounds on $(\mathbf{A}) - (\mathbf{D})$ yield for $N_h^t > 0$

$$\delta_h^t \le \left(1 + \frac{1}{H}\right)\delta_h^t + \left(1 + \frac{1}{H}\right)\xi_h^t$$

$$+ 3L\sqrt{\frac{\mathrm{Var}_{\overline{p}_h^{t-1}}[\overline{V}_{h+1}^{t-1}](s_h^t, a_h^t)}{\overline{N}_h^t}} + \sqrt{2L} \cdot \sqrt{\frac{\mathrm{Var}_{p_h}[V_{h+1}^\star](s_h^t, a_h^t)}{N_h^t}}$$

$$+ \frac{10H^2 S \cdot L}{N_h^t} + \frac{16L^2 H}{\overline{N}_h^t}.$$

Additionally, there is a trivial bound $\delta_h^t \le 2H$ that is valid for the case $N_h^t = 0$.

Define $\gamma_{h'} = (1 + 1/H)^{H-h'+1}$ and notice that $\gamma_{h'} \le \mathrm{e}, \overline{N}_h^t \ge N_h^t$. Summing it up in the definition of $\overline{\mathfrak{R}}_h^T$ we obtain

$$\overline{\mathfrak{R}}_h^T \le \sum_{t=1}^T \sum_{h'=h}^H \gamma_{h'}\xi_{h'}^t + \sum_{t=1}^T \sum_{h'=h}^H 2\mathrm{e}H\mathbb{1}\{N_{h'}^t = 0\}$$

$$+ 3\mathrm{e}L\sum_{t=1}^T \sum_{h'=h}^H \sqrt{\mathrm{Var}_{\overline{p}_{h'}^{t-1}}[\overline{V}_{h+1}^{t-1}](s_{h'}^t, a_{h'}^t) \cdot \frac{\mathbb{1}\{N_{h'}^t > 0\}}{N_{h'}^t}} \qquad \triangleq A_h^T$$

$$+ \mathrm{e}\sqrt{2L}\sum_{t=1}^T \sum_{h'=h}^H \sqrt{\mathrm{Var}_{p_{h'}}[V_{h'+1}^\star](s_{h'}^t, a_{h'}^t)\frac{\mathbb{1}\{N_{h'}^t > 0\}}{N_{h'}^t}} \qquad \triangleq B_h^T$$

$$+ 26H^2 SL^2\sum_{t=1}^T \sum_{h'=h}^H \frac{\mathbb{1}\{N_{h'}^t > 0\}}{N_{h'}^t} \qquad \triangleq C_h^T.$$

The bound on the first term is this decomposition follows from the definition of the event $\mathcal{E}(\delta) \subseteq \mathcal{G}^{\mathrm{conc}}(\delta) \subseteq \mathcal{G}(\delta)$. To bound the second term we notice that the event $\mathbb{1}\{N_{h'}^t = 0\}$ could occur no more than $SAH$ times. $\qquad \square$

Next we provide two important technical results. First of them is a classical result that follows from the pigeonhole principle.

**Lemma B.9.** *For any $H, T \geq 1$,*

$$\sum_{t=1}^{T} \sum_{h=1}^{H} \frac{\mathbb{1}\{n_h^{t-1}(s_h^t, a_h^t) > 0\}}{n_h^{t-1}(s_h^t, a_h^t)} \leq 2HSAL,$$

$$\sum_{t=1}^{T} \sum_{h=1}^{H} \frac{\mathbb{1}\{n_h^{t-1}(s_h^t, a_h^t) > 0\}}{\sqrt{n_h^{t-1}(s_h^t, a_h^t))}} \leq 3H\sqrt{TSA}.$$

*Proof.* The main observation for both inequalities follows from pigeon-hole principle: term corresponding to each state-action pair $(s, a)$ appears in the sum exactly $n_h^{t-1}(s, a)$ times with a value $1/n$ for $n$ increasing from 1, thanks to the indicator, to $n_h^{t-1}(s, a)$. For the first sum we use a bound on harmonic numbers, for the second one the integral bound. $\qquad\square$

**Lemma B.10.** *Assume that conditions of Theorem 3.1 are fulfilled. Then it holds on the event $\mathcal{G}(\delta)$,*

$$\sum_{t=1}^{T} \sum_{h=1}^{H} \mathrm{Var}_{\overline{p}_h^{t-1}}[\overline{V}_{h+1}^{t-1}](s_h^t, a_h^t)\mathbb{1}\{N_h^t > 0\} \leq 2H^2 T + 2H^2 U_1^T + 38H^3 S^2 AL^3 + 32H^3 S\sqrt{2ATL},$$

$$\sum_{t=1}^{T} \sum_{h=1}^{H} \mathrm{Var}_{p_h}[V_{h+1}^\star](s_h^t, a_h^t) \leq 2H^2 T + 2H^2 U_1^T + 6H^3 L + 8\sqrt{2H^5 TL}.$$

*where $U_h^T$ is defined in Lemma B.8.*

*Proof.* First, apply the second inequality in Lemma E.3,

$$\sum_{t=1}^{T} \sum_{h=1}^{H} \mathrm{Var}_{\overline{p}_h^{t-1}}[\overline{V}_{h+1}^{t-1}](s_h^t, a_h^t)\mathbb{1}\{N_h^t > 0\} \leq \underbrace{\sum_{t=1}^{T} \sum_{h=1}^{H} \mathrm{Var}_{p_h}[\overline{V}_{h+1}^{t-1}](s_h^t, a_h^t)\mathbb{1}\{N_h^t > 0\}}_{(\mathbf{W})}$$

$$+ \underbrace{2r_0^2 H^2 \sum_{t=1}^{T} \sum_{h=1}^{H} \|\overline{p}_h^{t-1}(s_h^t, a_h^t) - p_h(s_h^t, a_h^t)\|_1 \mathbb{1}\{N_h^t > 0\}}_{(\mathbf{X})}.$$

To bound the term $(\mathbf{X})$ one may use Lemma B.3. We obtain under assumption $N_h^t > 0$

$$\|\overline{p}_h^{t-1}(s_h^t, a_h^t) - p_h(s_h^t, a_h^t)\|_1 \leq \|\overline{p}_h^{t-1}(s_h^t, a_h^t) - \widehat{p}_h^{t-1}(s_h^t, a_h^t)\|_1 + \|p_h(s_h^t, a_h^t) - \widehat{p}_h^{t-1}(s_h^t, a_h^t)\|_1$$

$$\leq \frac{n_0}{N_h^t} + \sum_{s \in \mathcal{S}} N_h^t(s)\left(\frac{1}{N_h^t} - \frac{1}{\overline{N}_h^t}\right) + \sqrt{\frac{2SL}{N_h^t}} \leq \frac{2SL^2}{N_h^t} + \sqrt{\frac{2SL}{N_h^t}}.$$

Since $r_0 = 2$, Lemma B.9 implies

$$(\mathbf{X}) \leq 2r_0^2 H^2 \sum_{t=1}^{T} \sum_{h=1}^{H} \|\overline{p}_h^{t-1}(s_h^t, a_h^t) - p_h(s_h^t, a_h^t)\|_1 \mathbb{1}\{N_h^t > 0\} \leq 32H^3 S^2 AL^3 + 24H^3 S\sqrt{2ATL}.$$

Next, we bound $(\mathbf{W})$ using the first inequality of Lemma E.3

$$(\mathbf{W}) \leq \underbrace{2\sum_{t=1}^{T} \sum_{h=1}^{H} \mathrm{Var}_{p_h}[V_{h+1}^{\pi^t}](s_h^t, a_h^t)}_{(\mathbf{Y})} + \underbrace{2\sum_{t=1}^{T} \sum_{h=1}^{H} r_0 H p_h \left|\overline{V}_{h+1}^{t-1} - V_{h+1}^{\pi^t}\right|(s_h^t, a_h^t)}_{(\mathbf{Z})}.$$

The term $(\mathbf{Y})$ could be bounded using definition of the event $\mathcal{E}^{\mathrm{Var}}(\delta)$. It follows that

$$(\mathbf{Y}) \leq H^2 T + \sqrt{2H^5 TL} + 3H^3 L.$$

By Proposition B.6 we have $\overline{V}_{h+1}^{t-1}(s) \geq V_{h+1}^{\pi^t}(s)$ for any $s \in \mathcal{S}$. By the definition of $\xi_h^t, \delta_h^t$, definition of event $\mathcal{E}(\delta)$ term, and Lemma B.8 $(\mathbf{Z})$ could be bounded as follows

$$(\mathbf{Z}) \leq \sum_{t=1}^{T}\sum_{h=1}^{H} 2H(\xi_h^t + \delta_{h+1}^t) \leq 2r_0 H^2\sqrt{2TL} + 2H\sum_{h=1}^{H}\overline{\mathfrak{R}}_{h+1}^{T} \leq 4H^2\sqrt{2TL} + 2H^2 U_1^T.$$

Therefore, we have

$$\sum_{t=1}^{T}\sum_{h=1}^{H}\mathrm{Var}_{\overline{p}_h^{t-1}}[\overline{V}_{h+1}^{t-1}](s_h^t, a_h^t) \leq (\mathbf{X}) + 2\cdot(\mathbf{Y}) + 2\cdot(\mathbf{Z})$$
$$\leq 2H^2T + 2H^2U_1^T + (32+6)H^3S^2AL^3 + (24+8)H^3S\sqrt{2ATL}$$
$$\leq 2H^2T + 2H^2U_1^T + 38H^3S^2AL^3 + 32H^3S\sqrt{2ATL}.$$

The first inequality in Lemma E.3 gives us a bound for the second inequality

$$\sum_{t=1}^{T}\sum_{h=1}^{H}\mathrm{Var}_{p_h}[V_{h+1}^{\star}](s_h^t, a_h^t) \leq 2\underbrace{\sum_{t=1}^{T}\sum_{h=1}^{H}\mathrm{Var}_{p_h}[V_{h+1}^{\pi^t}](s_h^t, a_h^t)}_{(\mathbf{Y})} + 2\sum_{t=1}^{T}\sum_{h=1}^{H} r_0 H p_h\left|V_{h+1}^{\star} - V_{h+1}^{\pi^t}\right|(s_h^t, a_h^t).$$

Since $\overline{V}_h^{t-1} \geq V_h^{\star}$ by Proposition B.6, the second term is bounded by $(\mathbf{Z})$. Thus

$$\sum_{t=1}^{T}\sum_{h=1}^{H}\mathrm{Var}_{p_h}[V_{h+1}^{\star}](s_h^t, a_h^t) \leq 2(\mathbf{Y}) + 2(\mathbf{Z}) \leq 2H^2T + 2H^2U_1^T + 8\sqrt{2H^5TL} + 6H^3L.$$

$\square$

**Lemma B.11.** *Assume conditions of Theorem 3.1 and Lemma B.8. Then on the event $\mathcal{G}(\delta)$ it holds*

$$A_1^T \leq 6\mathrm{e}\sqrt{H^3SAT}\cdot L^{3/2} + 6\mathrm{e}\sqrt{H^3SAU_1^T}\cdot L^{3/2} + 27\mathrm{e}H^2S^{3/2}AL^3 + 30\mathrm{e}H^2SA^{3/4}T^{1/4}L^{7/4},$$
$$B_1^T \leq 4\mathrm{e}\sqrt{H^3SAT}\cdot L + 4\mathrm{e}\sqrt{H^3SAU_1^T}\cdot L + 8\mathrm{e}H^2S^{1/2}A^{1/2}L^2 + 10\mathrm{e}H^{7/4}S^{1/2}A^{1/2}T^{1/4}L^{5/4},$$
$$C_1^T \leq 52\mathrm{e}H^3S^2AL^3 = \widetilde{\mathcal{O}}(H^3S^2A).$$

*Proof.* For the term $A_1^T$ we apply the Cauchy—Schwartz inequality, Lemma B.10, Lemma B.9 and inequality $\sqrt{a+b} \leq \sqrt{a} + \sqrt{b}, a, b \geq 0$,

$$\sum_{t=1}^{T}\sum_{h=1}^{H}\sqrt{\mathrm{Var}_{\overline{p}_h^{t-1}}[\overline{V}_{h+1}^{t-1}](s_h^t, a_h^t)\frac{\mathbb{1}\{N_h^t > 0\}}{N_h^t}}$$
$$\leq \sqrt{\sum_{t=1}^{T}\sum_{h=1}^{H}\mathrm{Var}_{\overline{p}_h^{t-1}}[\overline{V}_{h+1}^{t-1}](s_h^t, a_h^t)\mathbb{1}\{N_h^t > 0\}}\cdot\sqrt{\sum_{t=1}^{T}\sum_{h=1}^{H}\frac{\mathbb{1}\{N_h^t > 0\}}{N_h^t}}$$
$$\leq \sqrt{2H^2T + 2H^2U_1^T + 38H^3S^2AL^3 + 32H^3S\sqrt{2ATL}}\cdot\sqrt{2SAHL}$$
$$\leq 2\sqrt{H^3SATL} + 2\sqrt{H^3SAU_1^TL} + 9H^2S^{3/2}AL^2 + 10H^2SA^{3/4}T^{1/4}L^{3/4}.$$

Similarly, the term $B_1^T$ may be estimated as follows

$$\sum_{t=1}^{T}\sum_{h=1}^{H}\sqrt{\mathrm{Var}_{p_h}[V_{h+1}^{\star}](s_h^t, a_h^t)\frac{\mathbb{1}\{N_h^t > 0\}}{N_h^t}} \leq \sqrt{\sum_{t=1}^{T}\sum_{h=1}^{H}\mathrm{Var}_{p_h}[V_{h+1}^{\star}](s_h^t, a_h^t)}\cdot\sqrt{\sum_{t=1}^{T}\sum_{h=1}^{H}\frac{\mathbb{1}\{N_h^t > 0\}}{N_h^t}}$$
$$\leq \sqrt{2H^2T + 2H^2U_1^T + 8\sqrt{2H^5TL} + 6H^3L}\cdot\sqrt{2SAH\cdot L}$$
$$\leq 2\sqrt{H^3SATL} + 2\sqrt{H^3SAU_1^TL} + 4H^2L\sqrt{SA} + 5H^{7/4}T^{1/4}L^{3/4}\sqrt{SA}.$$

Finally, to estimate $C_1^T$ we apply Lemma B.9. We obtain

$$C_1^T \le 26\mathrm{e}H^2 S \cdot L^2 \cdot 2SAHL \le 52\mathrm{e}H^3 S^2 AL^3.$$

$\square$

*Proof of Theorem 3.1.* By Corollary B.7 event $\mathcal{G}(\delta)$ holds with probability at least $1 - \delta$. Next we assume that this event holds. Then we have two cases: $T < H^2 S^2 AL^3$ and $T \ge H^2 S^2 AL^3$. In the first case the regret is trivially bounded by $\mathfrak{R}^T \le H^3 S^2 AL^3$. Thus it is sufficient to analyze only the second case.

By Proposition B.6 and Lemma B.8

$$\mathfrak{R}^T = \sum_{t=1}^{T} V_h^\star(s_1^t) - V_h^{\pi^t}(s_1^t) \le \sum_{t=1}^{T} \overline{V}_h^{t-1}(s_1^t) - V_h^{\pi^t}(s_1^t) = \overline{\mathfrak{R}}_1^T \tag{9}$$
$$\le U_1^T = A_1^T + B_1^T + C_1^T + 4\mathrm{e}\sqrt{2H^3 TL} + 2\mathrm{e}H^2 SA.$$

Next, under our condition on $T$ we can simplify expressions for the bounds of $A_1^T$ and $B_1^T$. Indeed, $T \ge H^2 S^2 AL^3$ implies that

$$H^{7/4} S^{1/2} A^{1/2} L^{5/4} \cdot T^{1/4} \le H^2 SA^{3/4} L^{7/4} \cdot T^{1/4} \le \sqrt{H^3 SAT} L^{3/2}.$$

Furthermore,

$$H^2 S^{3/2} AL^3 \le H^3 S^2 AL^3, \qquad H^2 S^{1/2} A^{1/2} L^2 \le H^3 S^2 AL^3, \qquad \sqrt{2H^3 TL} \le \sqrt{2H^3 SAT} \cdot L.$$

We obtain the following bounds

$$A_1^T \le 36\mathrm{e}\sqrt{H^3 SAT} \cdot L^{3/2} + 6\mathrm{e}\sqrt{H^3 SAU_1^T} \cdot L^{3/2} + 27\mathrm{e}H^3 S^2 AL^3,$$
$$B_1^T \le 14\mathrm{e}\sqrt{H^3 SAT} \cdot L + 4\mathrm{e}\sqrt{H^3 SAU_1^T} \cdot L + 8\mathrm{e}H^3 S^2 AL^3,$$
$$C_1^T \le 52\mathrm{e}H^3 S^2 AL^3.$$

Hence, using bound $H^2 SA \le H^3 S^2 A$,

$$U_1^T \le 50\mathrm{e}\sqrt{H^3 SAT} \cdot L^{3/2} + 10\mathrm{e}\sqrt{H^3 SAU_1^T} \cdot L^{3/2} + 89\mathrm{e}H^3 S^2 AL^3 + 4\mathrm{e}\sqrt{2} \cdot \sqrt{H^3 TL}$$
$$\le 56\mathrm{e}\sqrt{H^3 SAT} \cdot L^{3/2} + 10\mathrm{e}\sqrt{H^3 SAU_1^T} \cdot L^{3/2} + 89\mathrm{e}H^3 S^2 AL^3.$$

This is a quadratic inequality in $U_1^T$. Solving this inequality and using inequality $\sqrt{a + b} \le \sqrt{a} + \sqrt{b}, a, b \ge 0$, we obtain

$$U_1^T \le 108\mathrm{e}\sqrt{H^3 SATL^3} + 178\mathrm{e}H^3 S^2 AL^3 + 200\mathrm{e}^2 H^3 SAL^3.$$

The last inequality and (9) imply that

$$\mathfrak{R}^T = \mathcal{O}\Big(\sqrt{H^3 SATL^3} + H^3 S^2 AL^3\Big).$$

$\square$

## C Deviation inequalities

### C.1 Deviation inequality for categorical distributions

Next, we restate the deviation inequality for categorical distributions by Jonsson et al. [2020, Proposition 1]. Let $(X_t)_{t \in \mathbb{N}^\star}$ be i.i.d. samples from a distribution supported on $\{1, \dots, m\}$, of probabilities given by $p \in \Delta_{m-1}$, where $\Delta_{m-1}$ is the probability simplex of dimension $m-1$. We denote by $\widehat{p}_n$ the empirical vector of probabilities, i.e., for all $k \in \{1, \dots, m\}$,

$$\widehat{p}_{n,k} = \frac{1}{n} \sum_{\ell=1}^{n} \mathbb{1}\{X_\ell = k\}.$$

Note that an element $p \in \Delta_{m-1}$ can be seen as an element of $\mathbb{R}^{m-1}$ since $p_m = 1 - \sum_{k=1}^{m-1} p_k$. This will be clear from the context.

**Theorem C.1.** *For all $p \in \Delta_{m-1}$ and for all $\delta \in [0,1]$,*

$$\mathbb{P}(\exists n \in \mathbb{N}^\star, \, n \, \mathrm{KL}(\widehat{p}_n, p) > \log(1/\delta) + (m-1)\log(e(1 + n/(m-1)))) \leq \delta.$$

### C.2 Deviation inequality for categorical weighted sum

We fix a function $f : \{1, \dots, m\} \to [0, b]$ and recall the definition of the minimum Kullback-Leibler divergence for $p \in \Delta_{m-1}$ and $u \in \mathbb{R}$

$$\mathcal{K}_{\mathrm{inf}}(p, u, f) = \inf\{\mathrm{KL}(p, q) : q \in \Delta_{m-1}, qf \geq u\}.$$

As the Kullback-Leibler divergence this quantity admits a variational formula.

**Lemma C.2** (Lemma 18 by Garivier et al., 2018). *For all $p \in \Delta_{m-1}$, $u \in [0, b)$,*

$$\mathcal{K}_{inf}(p, u, f) = \max_{\lambda \in [0,1]} \mathbb{E}_{X \sim p}\left[\log\left(1 - \lambda \frac{f(X) - u}{b - u}\right)\right],$$

*moreover if we denote by $\lambda^\star$ the value at which the above maximum is reached, then*

$$\mathbb{E}_{X \sim p}\left[\frac{1}{1 - \lambda^\star \frac{f(X) - u}{b - u}}\right] \leq 1.$$

*Remark* C.3. Contrary to Garivier et al. [2018] we allow that $u = 0$ but in this case Lemma C.2 is trivially true, indeed

$$\mathcal{K}_{\mathrm{inf}}(p, 0, f) = 0 = \max_{\lambda \in [0,1]} \mathbb{E}_{X \sim p}\left[\log\left(1 - \lambda \frac{f(X)}{b}\right)\right].$$

We are now ready to restate the deviation inequality for the $\mathcal{K}_{\mathrm{inf}}$ by Tiapkin et al. [2022] which is a self-normalized version of Proposition 13 by Garivier et al. [2018].

**Theorem C.4.** *For all $p \in \Delta_{m-1}$ and for all $\delta \in [0, 1]$,*

$$\mathbb{P}\big(\exists n \in \mathbb{N}^\star, \, n \, \mathcal{K}_{inf}(\widehat{p}_n, pf, f) > \log(1/\delta) + 3\log(e\pi(1 + 2n))\big) \leq \delta.$$

### C.3 Deviation inequality for bounded distributions

Below, we restate the self-normalized Bernstein-type inequality by Domingues et al. [2020]. Let $(Y_t)_{t \in \mathbb{N}^\star}$, $(w_t)_{t \in \mathbb{N}^\star}$ be two sequences of random variables adapted to a filtration $(\mathcal{F}_t)_{t \in \mathbb{N}}$. We assume that the weights are in the unit interval $w_t \in [0, 1]$ and predictable, i.e. $\mathcal{F}_{t-1}$ measurable. We also assume that the random variables $Y_t$ are bounded $|Y_t| \leq b$ and centered $\mathbb{E}[Y_t | \mathcal{F}_{t-1}] = 0$. Consider the following quantities

$$S_t \triangleq \sum_{s=1}^{t} w_s Y_s, \quad V_t \triangleq \sum_{s=1}^{t} w_s^2 \cdot \mathbb{E}\big[Y_s^2 | \mathcal{F}_{s-1}\big], \quad \text{and} \quad W_t \triangleq \sum_{s=1}^{t} w_s$$

and let $h(x) \triangleq (x+1)\log(x+1) - x$ be the Cramér transform of a Poisson distribution of parameter 1.

**Theorem C.5** (Bernstein-type concentration inequality). *For all $\delta > 0$,*

$$\mathbb{P}\left(\exists t \geq 1, (V_t/b^2 + 1)h\left(\frac{b|S_t|}{V_t + b^2}\right) \geq \log(1/\delta) + \log(4e(2t+1))\right) \leq \delta.$$

*The previous inequality can be weakened to obtain a more explicit bound: if $b \geq 1$ with probability at least $1 - \delta$, for all $t \geq 1$,*

$$|S_t| \leq \sqrt{2V_t \log(4e(2t+1)/\delta)} + 3b\log(4e(2t+1)/\delta).$$

## C.4 Deviation inequality for Dirichlet distribution

Below we provide the Bernstein-type inequality for weighted sum of Dirichlet distribution, using a generalized result on upper bound on tails for linear statistics on Dirichlet distribution (Theorem D.1).

**Lemma C.6.** *[Generalization of Lemma C.8 by* Tiapkin et al. [2022]*] For any* $\alpha = (\alpha_0, \alpha_1, \ldots, \alpha_m) \in \mathbb{R}_{++}^{m+1}$ *define* $\overline{p} \in \Delta_m$ *such that* $\overline{p}(\ell) = \alpha_\ell/\overline{\alpha}, \ell = 0, \ldots, m$, *where* $\overline{\alpha} = \sum_{j=0}^m \alpha_j$. *Then for any* $f: \{0, \ldots, m\} \to [0, b]$ *such that* $f(0) = b$ *and* $\delta \in (0, 1)$

$$\mathbb{P}_{w \sim \mathcal{D}ir(\alpha)}\left[wf \geq \overline{p}f + 2\sqrt{\frac{\mathrm{Var}_{\overline{p}}(f)\log(1/\delta)}{\overline{\alpha}}} + \frac{3b \cdot \log(1/\delta)}{\overline{\alpha}}\right] \leq \delta.$$

*Remark* C.7. The only difference with the result of Lemma C.8 by Tiapkin et al. [2022] is allowing to parameters of Dirichlet distribution being non-integer.

*Proof.* Fix $\delta \in (0, 1)$ and let $\mu \in (\overline{p}f, b)$ be such that

$$\mathcal{K}_{\mathrm{inf}}(\overline{p}, \mu, f) = \overline{\alpha}^{-1}\log(1/\delta).$$

Note that such $\mu$ exists by the continuity of $\mathcal{K}_{\mathrm{inf}}$ w.r.t. the second argument, see Honda and Takemura [2010, Theorem 7]. By Tiapkin et al. [2022, Lemma C.7] there exists $q$ such that $\overline{p} \ll q$, $qf = \mu$ and $\mathrm{KL}(\overline{p}, q) = \overline{\alpha}^{-1}\log(1/\delta)$. By Theorem D.1

$$\mathbb{P}_{w \sim \mathcal{D}ir(\alpha)}[wf \geq qf] = \mathbb{P}_{w \sim \mathcal{D}ir(\alpha)}[wf \geq \mu] \leq \exp(-\overline{\alpha}\,\mathcal{K}_{\mathrm{inf}}(\overline{p}, \mu, f)) = \delta. \tag{10}$$

By Lemma E.1

$$qf - \overline{p}f \leq \sqrt{2\mathrm{Var}_q(f)\,\mathrm{KL}(\overline{p}, q)}.$$

By Lemma E.2, $\mathrm{Var}_q(f) \leq 2\mathrm{Var}_{\overline{p}}(f) + 4b^2\,\mathrm{KL}(\overline{p}, q)$. The last two inequalities and (10) imply that

$$\mathbb{P}_{w \sim \mathcal{D}ir(\alpha)}\left[wf - \overline{p}f \geq \sqrt{4\mathrm{Var}_{\overline{p}}(f)\,\mathrm{KL}(\overline{p}, q)} + 2b\sqrt{2} \cdot \mathrm{KL}(\overline{p}, q)\right] \leq \delta.$$

$\square$

# D  Exponential and Gaussian approximations of Dirichlet distribution

In this section we present result on approximation of a tail probabilities for linear statistics of Dirichlet distribution from above by tails of exponential distribution and from below by tails of Gaussian distribution.

The proof of upper bound generalizes proof of Baudry et al. [2021b] to non-integer parameters using exactly the same technique; see also Riou and Honda [2020].

**Theorem D.1** (Upper bound). *For any $\alpha = (\alpha_0, \alpha_1, \ldots, \alpha_m) \in \mathbb{R}_{++}^{m+1}$ define $\overline{p} \in \Delta_m$ such that $\overline{p}(\ell) = \alpha_\ell / \overline{\alpha}, \ell = 0, \ldots, m$, where $\overline{\alpha} = \sum_{j=0}^m \alpha_j$. Then for any $f \colon \{0, \ldots, m\} \to [0, b]$ and $0 < \mu < b$ and*

$$\mathbb{P}_{w \sim \mathcal{D}\mathrm{ir}(\alpha)}[wf \geq \mu] \leq \exp(-\overline{\alpha}\,\mathcal{K}_{\mathit{inf}}(\overline{p}, \mu, f)).$$

*Proof.* The statement is trivial for $\mu \leq \overline{p}f$ since $\mathcal{K}_{\inf}(\overline{p}, \mu, f) = 0$. Assume that $\mu > \overline{p}f$. It is well know fact that $w \sim \mathcal{D}\mathrm{ir}(\alpha)$ may be represented as follows

$$w \triangleq \left(\frac{Y_0}{V_m}, \frac{Y_1}{V_m}, \ldots, \frac{Y_m}{V_m}\right),$$

where $Y_\ell \overset{\mathrm{ind}}{\sim} \Gamma(\alpha_\ell, 1), \ell = 0, \ldots, m$ and $V_m = \sum_{\ell=0}^m Y_\ell$. Let us fix $\lambda \in [0, 1/(b - u))$ and proceed by the changing of measure argument

$$\mathbb{P}(wf \geq \mu) = \mathbb{P}\left(\sum_{\ell=0}^m Y_\ell f(\ell) \geq \sum_{\ell=0}^m Y_\ell \mu\right) = \mathbb{E}_{Y_\ell \overset{\mathrm{ind}}{\sim} \Gamma(\alpha_\ell, 1)}\left[\mathbb{1}\left\{\sum_{\ell=0}^m Y_\ell(f(\ell) - \mu) \geq 0\right\}\right]$$

$$= \mathbb{E}_{\hat{Y}_\ell \overset{\mathrm{ind}}{\sim} \Gamma(\alpha_\ell, 1 - \lambda(f(\ell) - \mu))}\left[\mathbb{1}\left\{\sum_{\ell=0}^m \hat{Y}_\ell(f(\ell) - \mu) \geq 0\right\} \prod_{\ell=0}^m \frac{\exp(-\lambda \hat{Y}_\ell(f(\ell) - \mu))}{(1 - \lambda(f(\ell) - \mu))^{\alpha_\ell}}\right]$$

$$= \exp\left(-\sum_{\ell=0}^m \alpha_\ell \log(1 - \lambda(f(\ell) - \mu))\right)$$

$$\cdot \mathbb{E}_{\hat{Y}_\ell \overset{\mathrm{ind}}{\sim} \Gamma(\alpha_\ell, 1 - \lambda(f(\ell) - \mu))}\left[\mathbb{1}\left\{\sum_{\ell=0}^m \hat{Y}_\ell(f(\ell) - \mu) \geq 0\right\} e^{-\lambda \sum_{\ell=0}^m \hat{Y}_\ell(f(\ell) - \mu)}\right]$$

$$\leq \exp\left(-\overline{\alpha}\sum_{\ell=0}^m \overline{p}_\ell \log(1 - \lambda(f(\ell) - \mu))\right) = \exp(-\overline{\alpha}\mathbb{E}_{X \sim \overline{p}}[\log(1 - \lambda(f(X) - \mu)]).$$

Since the previous inequality is true for all $\lambda \in [0, 1/(b - \mu))$, then the variational formula (Lemma C.2) allows to conclude

$$\mathbb{P}_{w \sim \mathcal{D}\mathrm{ir}(\alpha)}[wf \geq \mu] \leq \exp\left(-\overline{\alpha} \sup_{\lambda \in [0, 1/(b-\mu))} \mathbb{E}_{X \sim \overline{p}}[\log(1 - \lambda(f(X) - \mu)]\right)$$

$$= \exp(-\overline{\alpha}\,\mathcal{K}_{\inf}(\overline{p}, \mu, f)).$$

$\square$

The proof of lower bound extends the approach of Tiapkin et al. [2022] by ideas of Alfers and Dinges [1984] and gives much more exact bounds. Define

$$c_0(\varepsilon) = \left(\frac{4}{\sqrt{\log(17/16)}} + 8 + \frac{49 \cdot 4\sqrt{6}}{9}\right)^2 \frac{2}{\pi \cdot \varepsilon^2} + \log_{17/16}\left(\frac{5}{32 \cdot \varepsilon^2}\right). \qquad (11)$$

**Theorem D.2** (Lower bound). *For any $\alpha = (\alpha_0 + 1, \alpha_1, \ldots, \alpha_m) \in \mathbb{R}_{++}^{m+1}$ define $\overline{p} \in \Delta_m$ such that $\overline{p}(\ell) = \alpha_\ell / \overline{\alpha}, \ell = 0, \ldots, m$, where $\overline{\alpha} = \sum_{j=0}^m \alpha_j$. Let $\varepsilon \in (0, 1)$. Assume that $\alpha_0 \geq c_0(\varepsilon) + \log_{17/16}(\overline{\alpha})$ for $c_0(\varepsilon)$ defined in (11), and $\overline{\alpha} \geq 2\alpha_0$. Then for any $f \colon \{0, \ldots, m\} \to [0, b_0]$ such that $f(0) = b_0, f(j) \leq b < b_0/2, j \in \{1, \ldots, m\}$ and $\mu \in (\overline{p}f, b_0)$*

$$\mathbb{P}_{w \sim \mathcal{D}\mathrm{ir}(\alpha)}[wf \geq \mu] \geq (1 - \varepsilon)\mathbb{P}_{g \sim \mathcal{N}(0,1)}\left[g \geq \sqrt{2\overline{\alpha}\,\mathcal{K}_{\mathit{inf}}(\overline{p}, \mu, f)}\right].$$

In the further subsections we are going to prove this theorem.

## D.1  Proof of Theorem D.2

First, we restate the result of Tiapkin et al. [2022] on representation of the density of linear statistic of Dirichlet distribution.

**Proposition D.3** (Proposition D.3 of Tiapkin et al. [2022]). *Let $f \in \mathrm{F}_m(b)$ and $\alpha = (\alpha_0 + 1, \alpha_1, \ldots, \alpha_m) \in \mathbb{R}_{++}^{m+1}$ such that $\overline{\alpha} = \sum_{j=0}^{m} \alpha_j > 1$. Let $w \sim \mathcal{D}\mathrm{ir}(\alpha)$ and assume that $Z = wf$ is not degenerate. Then for any $0 \leq u < b_0$*

$$p_Z(u) = \frac{\overline{\alpha}}{2\pi} \int_{\mathbb{R}} (1 + \mathrm{i}(b_0 - u)s)^{-1} \prod_{j=0}^{m} (1 + \mathrm{i}(f(j) - u)s)^{-\alpha_j} \mathrm{d}s.$$

Next we proceed in the same spirit as an approach of Tiapkin et al. [2022] and apply the method of saddle point (see Fedoryuk [1977], Olver [1997]) to derive an asymptotically tight approximation. However, in our case we have to extract one additional term in from of the product.

**Proposition D.4.** *Let $f \in \mathrm{F}_m(b_0, b)$ and let $\alpha = (\alpha_0 + 1, \alpha_1, \ldots, \alpha_m) \in \mathbb{R}_+^{m+1}$ be a fixed vector with $\alpha_0 \geq 2$. Then for any $u \in (\overline{p}f, b_0)$,*

$$\int_{\mathbb{R}} \frac{\prod_{\ell=0}^{m}(1 + \mathrm{i}(f(\ell) - u)s)^{-\alpha_\ell}}{(1 + \mathrm{i}(b_0 - u)s)} \mathrm{d}s = \left( \sqrt{\frac{2\pi}{\overline{\alpha}\,\sigma^2}} - R_1(\alpha) + R_2(\alpha) \right) \frac{\exp(-\overline{\alpha}\,\mathcal{K}_{inf}(\overline{p}, u, f))}{1 - \lambda^\star(b_0 - u)} + R_3(\alpha),$$

*where*

$$\sigma^2 = \mathbb{E}_{X \sim \overline{p}} \left[ \left( \frac{f(X) - u}{1 - \lambda^\star(f(X) - u)} \right)^2 \right],$$

$$|R_1(\alpha)| \leq \frac{c_1}{(1 - \lambda^\star(b_0 - u))\sqrt{\sigma^2\,c_\kappa \alpha_0\,\overline{\alpha}}},$$

$$|R_2(\alpha)| \leq \frac{c_2}{(1 - \lambda^\star(b_0 - u))\sqrt{\sigma^2\,\overline{\alpha}\,\alpha_0}},$$

$$|R_3(\alpha)| \leq c_3 \cdot \frac{\exp(-\overline{\alpha}\,\mathcal{K}_{inf}(\overline{p}, u, f))}{1 - \lambda^\star(b_0 - u)} \cdot \frac{1 - \lambda^\star(b_0 - u)}{b_0 - u} \exp(-c_\kappa \alpha_0)$$

*with $c_1 = 2\sqrt{2}, c_2 = \left( 8 + \frac{49\sqrt{6}}{9} \frac{b_0}{b_0 - \overline{p}f} \right), c_3 = \frac{\sqrt{5\pi}}{2}, c_\kappa = 1/2 \cdot \log\left( 1 + \frac{1}{4}\left( \frac{b_0 - \overline{p}f}{b_0} \right)^2 \right)$ and $\lambda^\star$ being a solution to the optimization problem*

$$\lambda^\star(\overline{p}, u, f) = \operatorname*{arg\,max}_{\lambda \in [0, 1/(b_0 - u)]} \mathbb{E}_{X \sim \overline{p}}[\log(1 - \lambda(f(X) - u))].$$

*Proof.* We start from the rewriting the integral in the form that allows us to apply saddle point method,

$$I = \int_{\mathbb{R}} \frac{\prod_{j=0}^{n}(1 + \mathrm{i}(f(j) - u)s)^{-\alpha_j}}{1 - \mathrm{i}(b_0 - u)s} \mathrm{d}s = \int_{\mathbb{R}} \frac{\exp\left( -\overline{\alpha}\sum_{j=0}^{m} \overline{p}_j \log(1 + \mathrm{i}(f(j) - u)s) \right)}{1 - \mathrm{i}(b_0 - u)s} \mathrm{d}s$$

$$= \int_{\mathbb{R}} (1 - \mathrm{i}(b_0 - u)s)^{-1} \exp(-\overline{\alpha}\,\mathbb{E}_{X \sim \overline{p}}[\log(1 + \mathrm{i}(f(X) - u)s)]) \mathrm{d}s. \tag{12}$$

Since the analysis of the suitable integration contour depends only on the function under exponent, we may directly switch to the contour $\gamma^\star = \mathrm{i}\lambda^\star + \mathbb{R}$ as it was stated in Tiapkin et al. [2022].

Next we continue following approach of Tiapkin et al. [2022] and denote the following functions

$$T(s) = \mathbb{E}[\log(1 - \lambda^\star(f(X) - u) + \mathrm{i}s(f(X) - u))],$$

$$P(s) = \frac{1}{1 - \lambda^\star(b_0 - u) + \mathrm{i}s(b_0 - u)},$$

a cut-off parameter $K > 0$, and define $\kappa_1 = T(-K) - T(0)$, $\kappa_2 = T(K) - T(0)$. Similarly to Chapter 4 (Section 6) by Olver [1997], we define the change of variables $v_1 = T(-s) - T(0)$, $v_2 = T(s) - T(0)$ and the implicit functions $q_1(v_1) = \frac{P(-s)}{T'(-s)}$, $q_2(v_2) = \frac{P(s)}{T'(s)}$. Notice that these functions differs from ones defined in Tiapkin et al. [2022] due to the presence of an additional multiplier $P(s)$. Using the first order Taylor expansion, we can write $q_1(v_1) = \frac{P(0)}{\sqrt{2T''(0)\cdot v_1}} + r_1(v_1)$, $q_2(v_2) = \frac{P(0)}{\sqrt{2T''(0)\cdot v_2}} + r_2(v_2)$. Then we have the following decomposition

$$I = \int_{-\infty}^{\infty} P(s) \exp(-\overline{\alpha}\, T(s))\, \mathrm{d}s = \left( P(0) \cdot \sqrt{\frac{2\pi}{\overline{\alpha}\, T''(0)}} - R_1(\alpha) + R_2(\alpha) \right) \exp(-\overline{\alpha}\, T(0)) + R_3(\alpha),$$

where

$$R_1(\alpha) = \left( \Gamma\left(\frac{1}{2}, \kappa_1\, \overline{\alpha}\right) + \Gamma\left(\frac{1}{2}, \kappa_2\, \overline{\alpha}\right) \right) \frac{P(0)}{\sqrt{2T''(0)\, \overline{\alpha}}},$$

$$R_2(\alpha) = \int_0^{\kappa_1} e^{-\overline{\alpha} v_1} r_1(v_1)\, \mathrm{d}v_1 + \int_0^{\kappa_2} e^{-\overline{\alpha} v_2} r_2(v_2)\, \mathrm{d}v_2,$$

$$R_3(\alpha) = \int_{\mathbb{R}\setminus[-K,K]} P(s) \exp(-\overline{\alpha}\, T(s))\, \mathrm{d}s,$$

where $\Gamma(\alpha, x)$ is an upper incomplete gamma function and integration w.r.t. $v_1, v_2$ is performed over the straight lines connecting the points $0$ and $\kappa_1, \kappa_2$, respectively. Define $\sigma^2 = T''(0)$.

**Term $R_2$.** We will start from upper bounding on remainder terms in Taylor-like expansions $r_2(v)$

$$|r_2(v)| = \left| \frac{P(s)}{T'(s)} - \frac{P(0)}{\sqrt{2T''(0)(T(s) - T(0))}} \right|$$

$$\leq P(0) \left| \frac{1}{T'(s)} - \frac{1}{\sqrt{2T''(0)(T(s) - T(0))}} \right| + \frac{|P(s) - P(0)|}{|T'(s)|}$$

$$= P(0) \cdot \overline{r}_2(v) + \tilde{r}_2(v).$$

Analysis of the term $\overline{r}_2(v)$ was performed in Tiapkin et al. [2022] under the choice $1/(2K) = \max\left\{ \frac{b_0 - u}{1 - \lambda^\star(b_0 - u)}, \frac{u}{1 + \lambda^\star u} \right\}$ and the upper bound $\kappa = \operatorname{Re}\kappa_2 = \operatorname{Re}\kappa_1 \geq c_\kappa \cdot \frac{\alpha_0}{\alpha}$ with $c_\kappa = 1/2 \cdot \log\left( 1 + \frac{1}{4}\left( \frac{b_0 - \overline{p}f}{b_0} \right)^2 \right)$ led to

$$\overline{r}_2(v) \leq \frac{49\sqrt{6}}{36\sqrt{\sigma^2}} \cdot \sqrt{\frac{\overline{\alpha}}{\alpha_0}} \frac{b_0}{b_0 - \overline{p}f}.$$

Our next goal is to analyze the second term $\tilde{r}_2(v)$. We apply Taylor expansions of the form $T'(s) = T''(0)s + T'''(\xi_2)s^2/2$ and $P(s) = P(0) + P'(\eta)s$ to derive

$$\tilde{r}_2(v) = \left| \frac{P(s) - P(0)}{T'(s)} \right| = \frac{|P'(\eta) \cdot s|}{|T''(0)s + T'''(\xi_2)s^2/2|} \leq \frac{\sup_{\eta \in (0,s)} |P'(\eta)|}{|T''(0) + T'''(\xi_2)s/2|}.$$

First note that $P'(\eta)$ maximizes at $\eta = 0$, since

$$P'(\eta) = \frac{b_0 - u}{(1 - \lambda^\star(b_0 - u) + i\eta(b_0 - u))^2}.$$

Next by defining a random variable $Y_s = \frac{f(X) - u}{1 - \lambda^\star(f(X) - u) + is(f(X) - u)}$ and due to our choice of $K$ we conclude that

$$|T''(0) + T'''(\xi_2)s/2| \geq \mathbb{E}[Y_0^2] - s\mathbb{E}[|Y_0|^3] \geq \mathbb{E}[Y_0^2]/2 = \sigma^2/2.$$

It yields

$$\tilde{r}_2(v) \leq \frac{2(b_0 - u)}{(1 - \lambda^\star(b_0 - u))^2 \sigma^2} = \frac{2}{(1 - \lambda^\star(b_0 - u))\sqrt{\sigma^2}} \sqrt{\frac{\frac{(b_0 - u)^2}{(1 - \lambda^\star(b_0 - u))^2}}{\mathbb{E}[Y_0^2]}}.$$

By a bound

$$\mathbb{E}[Y_0^2] = \sum_{i=0}^{m} \frac{\alpha_i}{\overline{\alpha}} \cdot \left( \frac{f(i) - u}{1 - \lambda^\star(f(i) - u)} \right)^2 \geq \frac{\alpha_0}{\overline{\alpha}} \frac{(b_0 - u)^2}{(1 - \lambda^\star(b_0 - u))^2}$$

we obtain

$$\tilde{r}_2(v) \leq \frac{2}{(1 - \lambda^\star(b_0 - u))\sqrt{\sigma^2}} \sqrt{\frac{\overline{\alpha}}{\alpha_0}}$$

and

$$|r_2(v)| \leq \frac{1}{(1 - \lambda^\star(b_0 - u))\sqrt{\sigma^2}} \sqrt{\frac{\overline{\alpha}}{\alpha_0}} \left( 2 + \frac{49\sqrt{6}}{36} \frac{b_0}{b_0 - \overline{p}f} \right).$$

A similar bound also holds for $r_1(v)$ by symmetry. Finally, due to bound on $\kappa$ and $\alpha_0 \geq 2$, we derive

$$|R_2(\alpha)| \leq \frac{2}{(1 - \lambda^\star(b_0 - u))\sqrt{\sigma^2}} \sqrt{\frac{\overline{\alpha}}{\alpha_0}} \left( 2 + \frac{49\sqrt{6}}{36} \frac{b_0}{b_0 - \overline{p}f} \right) \cdot \left| \int_0^{\kappa_2} e^{-\overline{\alpha}v} \mathrm{d}v + \int_0^{\kappa_1} e^{-\overline{\alpha}v} \mathrm{d}v \right|$$

$$\leq \frac{1}{(1 - \lambda^\star(b_0 - u))\sqrt{\sigma^2}} \left( 8 + \frac{49\sqrt{6}}{9} \frac{b_0}{b_0 - \overline{p}f} \right) \cdot \frac{1}{\sqrt{\overline{\alpha} \cdot \alpha_0}}.$$

**Term $R_1$.** The analysis of this term can be carried out as in Tiapkin et al. [2022] except the multiplication with $P(0)$,

$$|R_1(\alpha)| \leq \frac{c_1}{\sqrt{\sigma^2 c_\kappa \alpha_0} \cdot (1 - \lambda^\star(b_0 - u))} \cdot \frac{\exp(-c_\kappa \alpha_0)}{\overline{\alpha}^{1/2}},$$

where $c_1 = 2\sqrt{2}$.

**Term $R_3$.** We start from the bound

$$\left| \int_K^\infty P(s) \exp(-\overline{\alpha}T(s)) \, \mathrm{d}s \right| \leq \exp(-\overline{\alpha} \cdot \mathrm{Re}[T(K) - T(0)]) \cdot \exp(-\overline{\alpha}T(0))$$

$$\cdot \sup_{s \in \mathbb{R}} |P(s)| \int_K^\infty \exp(-\overline{\alpha} \, \mathrm{Re}[T(s) - T(K)]) \, \mathrm{d}s.$$

Let us start from the analysis of an additional multiplier connected to $P(s)$

$$\sup_s |P(s)| = \sup_s \sqrt{\frac{1}{(1 - \lambda^\star(b_0 - u))^2 + s^2(b_0 - u)}} = \frac{1}{1 - \lambda^\star(b_0 - u)}.$$

The rest of the analysis coincides the the analysis of the same term in Tiapkin et al. [2022]

$$|R_3(\alpha)| \leq c_3 \cdot \frac{\exp(-\overline{\alpha} \mathcal{K}_{\mathrm{inf}}(\overline{p}, u, f))}{1 - \lambda^\star(b_0 - u)} \cdot \frac{1 - \lambda^\star(b_0 - u)}{b_0 - u} \exp(-c_\kappa \alpha_0)$$

for $c_3 = \sqrt{5\pi}/2$. □

Finally, we use a bounds on remainder terms to derive a lower bound on the density.

**Lemma D.5.** *Consider a function $f \in \mathrm{F}_m(b_0, b)$ and a vector $\alpha = (\alpha_0 + 1, \alpha_1, \ldots, \alpha_m) \in \mathbb{R}_+^{m+1}$ with $\overline{\alpha} \geq 2\alpha_0$, $b_0 \geq 2b$. Let $w \sim \mathrm{Dir}(\alpha)$ and assume that $Z = wf$ is non-degenerate. Let $\varepsilon \in (0, 1)$. Assume*

$$\alpha_0 \geq \left( \frac{4}{\sqrt{\log(17/16)}} + 8 + \frac{49 \cdot 4\sqrt{6}}{9} \right)^2 \frac{2}{\pi \cdot \varepsilon^2} + \log_{17/16} \left( \frac{5}{32 \cdot \varepsilon^2} \right) + \log_{17/16}(\overline{\alpha}).$$

*Then for any $u \in (\overline{p}f, b_0)$,*

$$p_Z(u) \geq (1 - \varepsilon) \sqrt{\frac{\overline{\alpha}}{2\pi}} \frac{\exp(-\overline{\alpha} \mathcal{K}_{inf}(\overline{p}, u, f))}{(1 - \lambda^\star(b_0 - u))\sqrt{\sigma^2}}.$$

*Proof.* We start the proof from the combination of Proposition D.3 and Proposition D.4

$$p_Z(u) \geq \frac{\overline{\alpha}}{2\pi} \left( \left( \sqrt{2\pi} - \frac{1}{\sqrt{\alpha_0}} \left( \frac{c_1}{\sqrt{c_\kappa}} + c_2 \right) \right) \frac{\exp(-\overline{\alpha} \, \mathcal{K}_{\inf}(\overline{p}, u, f))}{(1 - \lambda^\star(b_0 - u))\sqrt{\overline{\alpha} \cdot \sigma^2}} - |R_3(\alpha)| \right).$$

Since $\overline{\alpha} \geq 2\alpha_0$ and $b_0 \geq 2b$ we have $b_0/(b_0 - \overline{p}f) \leq 4$. In this case we have $c_\kappa \geq 1/2 \log(17/16)$ and $c_2 \leq 8 + 49\sqrt{6} \cdot 4/9$. Therefore

$$\frac{c_1}{\sqrt{c_\kappa}} + c_2 \leq \frac{4}{\sqrt{\log(17/16)}} + 8 + \frac{49 \cdot 4\sqrt{6}}{9} \triangleq \gamma_1.$$

And for $\alpha_0 \geq 4\gamma_1^2/(2\pi \cdot \varepsilon^2)$, we have

$$p_Z(u) \geq \frac{\overline{\alpha}}{2\pi} \left( \sqrt{2\pi}(1 - \varepsilon/2) \frac{\exp(-\overline{\alpha} \, \mathcal{K}_{\inf}(\overline{p}, u, f)}{(1 - \lambda^\star(b_0 - u))\sqrt{\overline{\alpha} \cdot \sigma^2}} - |R_3(\alpha)| \right)$$

$$\geq \frac{\sqrt{\overline{\alpha}}}{2\pi} \left( \frac{\sqrt{2\pi}(1 - \varepsilon/2)}{\sqrt{\overline{\alpha}} \, \sigma^2} - c_3 \cdot \frac{1 - \lambda^\star(b_0 - u)}{b_0 - u} \cdot \exp(-c_\kappa \alpha_0) \right) \frac{\exp(-\overline{\alpha} \, \mathcal{K}_{\inf}(\overline{p}, u, f)}{(1 - \lambda^\star(b_0 - u))}.$$

Note that $\mathbb{E}[Y_0] = 0$ and observe that the inequality

$$\sigma^2 = \mathbb{E}[Y_0^2] = \text{Var}[Y_0] \leq \left( \frac{b_0 - u}{2(1 - \lambda^\star(b_0 - u))} + \frac{u}{2(1 + \lambda^\star u)} \right)^2$$

$$= \frac{b_0^2}{4(1 - \lambda^\star(b_0 - u))^2(1 + \lambda^\star u)^2} \leq \frac{4(b_0 - u)^2}{(1 - \lambda^\star(b_0 - u))^2},$$

follows from a general bound on variance of bounded random variables (bounded differences), the fact (see Lemma 12 in [Honda and Takemura [2010]](#))

$$\lambda^\star \geq \frac{u - \overline{p}f}{u(b_0 - u)} \iff 1 + \lambda^\star u \geq \frac{b_0 - \overline{p}f}{b_0 - u},$$

and the inequality $b_0/(b_0 - \overline{p}f) \leq 4$. It yields

$$p_Z(u) \geq \frac{\sqrt{\overline{\alpha}}}{2\pi} \left( \sqrt{2\pi}(1 - \varepsilon/2) - 2\sqrt{5\pi} \exp(-c_\kappa \alpha_0) \cdot \sqrt{\overline{\alpha}} \right) \frac{\exp(-\overline{\alpha} \, \mathcal{K}_{\inf}(\overline{p}, u, f))}{(1 - \lambda^\star(b_0 - u)) \cdot \sqrt{\sigma^2}}.$$

To guarantee

$$2\sqrt{5\pi} \exp(-c_\kappa \alpha_0) \cdot \sqrt{\overline{\alpha}} \leq \sqrt{2\pi} \cdot (\varepsilon/2)$$

we have to choose

$$\alpha_0 \geq \log_{17/16}(5/(32\varepsilon^2)) + \log_{17/16}(\overline{\alpha}).$$

It allows us to conclude

$$p_Z(u) \geq (1 - \varepsilon)\sqrt{\frac{\overline{\alpha}}{2\pi}} \frac{\exp(-\overline{\alpha} \, \mathcal{K}_{\inf}(\overline{p}, u, f))}{(1 - \lambda^\star(b_0 - u)) \cdot \sqrt{\sigma^2}}.$$

$\square$

Before proceeding with the final proof, we derive one important technical result.

**Lemma D.6.** *For any $u \in (\overline{p}f, b_0)$ it holds*

$$\mathcal{K}_{inf}(\overline{p}, u, f) \geq \frac{1}{2}(\lambda^\star)^2 \sigma^2 \left( 1 - \lambda^\star(b_0 - u) \right)^2.$$

*Proof.* Define the function $\phi_u(\lambda) = \mathbb{E} \log \left( 1 - \lambda(f(X) - u) \right)$ and $\lambda_u = \lambda^\star$. Remark that $\sigma^2 = -\phi_u''(\lambda_u)$. Thanks to the Taylor expansion of $\phi_u$ and the definition of $\lambda_u$ it holds

$$0 = \phi_u(0) = \phi_u(\lambda_u) + 0 + \frac{\lambda_u^2}{2} \phi_u''(y\lambda_u)$$

for some $y \in (0,1)$. Thus we can rewrite $\mathcal{K}_{\inf}$ as

$$\phi_u(\lambda) = \frac{\lambda_u^2}{2}\left(-\phi_u''(y\lambda_u)\right).$$

We will lower bound the opposite of the second derivative that appears above. First note that

$$-\phi_u''(y\lambda_u) = \mathbb{E}\left[\frac{(f(X)-u)^2}{\left(1-\lambda_u(f(X)-u)\right)^2}\left(\frac{1-\lambda_u(f(X)-u)}{1-y\lambda_u(f(X)-u)}\right)^2\right].$$

We now lower-bound the ratio, noting that if $X \leq u$ then since $y \in (0,1)$

$$\frac{1-\lambda_u(f(X)-u)}{1-y\lambda_u(f(X)-u)} \geq 1.$$

In the other case $X > u$, we have $0 \leq 1-y\lambda_u(f(X)-u) \leq 1$ and $1-\lambda_u(f(X)-u) \geq 1-\lambda_u(b_0-u)$ thus

$$\frac{1-\lambda_u(f(X)-u)}{1-y\lambda_u(f(X)-u)} \geq 1-\lambda_u(b-u) > 0.$$

In both case using $1-\lambda_u(b_0-u) \leq 1$ we get

$$\left(\frac{1-\lambda_u(f(X)-u)}{1-y\lambda_u(f(X)-u)}\right)^2 \geq \left(1-\lambda_u(b_0-u)\right)^2.$$

In particular, using the definition of $\phi''(u)$, it entails that

$$-\phi_u''(y\lambda_u) \geq -\phi_u''(\lambda_u)\left(1-\lambda_u(b-u)\right)^2.$$

Plugging this inequality in the integral representation of $\phi_u$ allows us to conclude

$$\phi_u(\lambda) \geq \frac{1}{2}\lambda_u^2\left(-\phi''(\lambda_u)\right)\left(1-\lambda_u(b-u)\right)^2.$$

$\square$

Using this lemma we may proceed with the proof of our final result.

*Proof of Theorem D.2.* Define $Z = wf$. By Lemma D.5,

$$\mathbb{P}(wf \geq \mu) = \int_\mu^{b_0} p_Z(u)\mathrm{d}u \geq (1-\varepsilon)\sqrt{\frac{\overline{\alpha}}{2\pi}} \cdot \int_\mu^{b_0} \frac{\exp(-\overline{\alpha}\,\mathcal{K}_{\inf}(\overline{p},u,f))}{\sqrt{\sigma^2(1-\lambda^\star(b_0-u))^2}}\,\mathrm{d}u.$$

By Theorem 6 by Honda and Takemura [2010],

$$\frac{\partial}{\partial u}\mathcal{K}_{\inf}(\overline{p},u,f) = \lambda^\star.$$

Thus, we can define a change of variables $t^2/2 = \mathcal{K}_{\inf}(\overline{p},u,f), t\mathrm{d}t = \lambda^\star\mathrm{d}u$ and write

$$\mathbb{P}(Z \geq \mu) = (1-\varepsilon)\int_{\sqrt{2\,\mathcal{K}_{\inf}(\overline{p},\mu,f)}}^{+\infty} D(u)\sqrt{\frac{\overline{\alpha}}{2\pi}}\exp(-\overline{\alpha}t^2/2)\mathrm{d}t,$$

where $D(u)$ is defined as a positive square root of

$$D^2(u) = \frac{2\,\mathcal{K}_{\inf}(\overline{p},u,f)}{(\lambda^\star)^2\sigma^2(1-\lambda^\star(b_0-u))^2}.$$

By Lemma D.6, $D^2(u) \geq 1$ and hence

$$\mathbb{P}(Z \geq \mu) \geq (1-\varepsilon)\int_{\sqrt{2\,\mathcal{K}_{\inf}(\overline{p},\mu,f)}}^{+\infty} \sqrt{\frac{\overline{\alpha}}{2\pi}}\exp(-\overline{\alpha}t^2/2)\mathrm{d}t$$

$$= (1-\varepsilon)\mathbb{P}_{g\sim\mathcal{N}(0,1)}\left(g \geq \sqrt{2\overline{\alpha}\,\mathcal{K}_{\inf}(\overline{p},\mu,f)}\right).$$

$\square$

# E   Technical lemmas

## E.1   On the Bernstein inequality

In this part, we restate Bernstein-type inequality of Talebi and Maillard [2018].

**Lemma E.1** (Corollary 11 by Talebi and Maillard, 2018)**.** *Let $p, q \in \Delta_{S-1}$, where $\Delta_{S-1}$ denotes the probability simplex of dimension $S - 1$. For all functions $f : \mathcal{S} \mapsto [0, b]$ defined on $\mathcal{S}$,*

$$pf - qf \leq \sqrt{2\mathrm{Var}_q(f)\,\mathrm{KL}(p, q)} + \frac{2}{3}b\,\mathrm{KL}(p, q)$$

$$qf - pf \leq \sqrt{2\mathrm{Var}_q(f)\,\mathrm{KL}(p, q)}\,.$$

*where use the expectation operator defined as $pf \triangleq \mathbb{E}_{s\sim p}f(s)$ and the variance operator defined as $\mathrm{Var}_p(f) \triangleq \mathbb{E}_{s\sim p}\big(f(s) - \mathbb{E}_{s'\sim p}f(s')\big)^2 = p(f - pf)^2$.*

**Lemma E.2** (Lemma E.3 by Tiapkin et al., 2022)**.** *Let $p, q \in \Delta_{S-1}$ and a function $f : \mathcal{S} \mapsto [0, b]$, then*

$$\mathrm{Var}_q(f) \leq 2\mathrm{Var}_p(f) + 4b^2\,\mathrm{KL}(p, q)\,,$$

$$\mathrm{Var}_p(f) \leq 2\mathrm{Var}_q(f) + 4b^2\,\mathrm{KL}(p, q)\,.$$

**Lemma E.3** (Lemma E.4 by Tiapkin et al., 2022)**.** *For $p, q \in \Delta_{S-1}$, for $f, g : \mathcal{S} \mapsto [0, b]$ two functions defined on $\mathcal{S}$, we have that*

$$\mathrm{Var}_p(f) \leq 2\mathrm{Var}_p(g) + 2bp|f - g| \quad \text{and}$$

$$\mathrm{Var}_q(f) \leq \mathrm{Var}_p(f) + 3b^2\|p - q\|_1,$$

*where we denote the absolute operator by $|f|(s) = |f(s)|$ for all $s \in \mathcal{S}$.*

# F  Lazy version of `OPSRL`

In this section we present `Lazy-OPSRL` a lazy version of the `OPSRL` algorithm. Following Efroni et al. [2019], instead of computing new Q-values by backward induction before each episode in `Lazy-OPSRL` we just just do one step of optimistic incremental planning at the current state to obtain improved Q-values (at the current state) and act greedily with respect to them. Precisely, given initial optimistic value functions $\overline{V}_h^{-1}(s) = r_0 H$ for all $(h,s) \in [H] \times \mathcal{S}'$ and Q-function $\overline{Q}_h^{-1}(s,a) = r_0 H$ for all $(h,s,a) \in [H] \times \mathcal{S}' \times \mathcal{A}$ we update Q-values by applying the Bellman operator *only at the visited states*:

$$\overline{Q}_h^t(s,a) \triangleq \mathbb{1}\{s = s_h^{t+1}\}\left(r_h(s,a) + \max_{j \in [J]}\{\widetilde{p}_h^{t,j}\overline{V}_{h+1}^{t-1}(s,a)\}\right) + (1 - \mathbb{1}\{s = s_h^{t+1}\})\overline{Q}_h^{t-1}(s,a),$$

$$\overline{V}_h^t(s) \triangleq \min\left\{\max_{a \in \mathcal{A}}\overline{Q}_h^t(s,a), \overline{V}_h^{t-1}(s)\right\},$$

$$\pi_h^{t+1}(s) \in \arg\max_{a \in \mathcal{A}}\overline{Q}_h^t(s,a),$$

(13)

where the posterior sample are still given by $\widetilde{p}_h^{t,j}(s,a) \sim \mathcal{D}ir\left(\left(\overline{n}_h^t(s'|s,a)/\kappa\right)_{s' \in \mathcal{S}'}\right)$ and $\overline{V}_{H+1}^t(s) = 0$ for all $t, s$. Consequently `Lazy-OPSRL` enjoys a better time complexity of $\widetilde{\mathcal{O}}(HSA)$ per episode than the one $\widetilde{\mathcal{O}}(HS^2A)$ of `OPSRL`.

The complete description of `Lazy-OPSRL` is given in Algorithm 2 for a general family of probability distribution parameterized by the pseudo-counts over the transitions instead of the Dirichlet inflated prior/posterior.

---

**Algorithm 2** `Lazy-OPSRL`

---

1: **Input:** Family of probability distributions $\rho : \mathbb{N}_+^{S+1} \to \Delta_{\mathcal{S}'}$ over transitions, initial pseudo-count $\overline{n}_h^0$, number of posterior samples $J$, initial value functions $\overline{V}_h^{-1}$, initial Q-functions $\overline{Q}_h^{-1}$.
2: **for** $t \in [T]$ **do**
3:    **for** $h \in [H]$ **do**
4:       Sample $J$ independent transitions $\widetilde{p}_h^{t-1,j}(s,a) \sim \rho(\overline{n}_h^{t-1}(s'|s,a)_{s' \in \mathcal{S}'}), \quad j \in [J]$.
5:       Compute for all $a \in \mathcal{A}$

$$\overline{Q}_h^{t-1}(s_h^t,a) = r_h(s_h^t,a) + \max_{j \in [J]}\{\widetilde{p}_h^{t-1,j}\overline{V}_{h+1}^{t-2}(s_h^t,a)\},$$

$$\overline{V}_h^{t-1}(s_h^t) = \min\left\{\max_{a \in \mathcal{A}}\overline{Q}_h^{t-1}(s_h^t,a), \overline{V}_h^{t-2}(s_h^t)\right\}.$$

6:       Play $a_h^t \in \arg\max_{a \in \mathcal{A}}\overline{Q}_h^{t-1}(s_h^t,a)$.
7:       Observe $s_{h+1}^t \sim p_h(s_h^t, a_h^t)$.
8:       Increment the pseudo-count $\overline{n}_h^t(s_{h+1}^t|s_h^t, a_h^t)$.
9:    **end for**
10: **end for**

---

Interestingly, we can also obtain a regret bound for `Lazy-OPSRL` of the same order as `OPSRL` with the same number of posterior samples as in 3.1.

**Theorem F.1.** *Consider a parameter* $\delta \in (0,1)$. *Let* $\kappa \triangleq 2(\log(12SAH/\delta) + 3\log(e\pi(2T + 1)))$, $n_0 \triangleq \lceil\kappa(c_0 + \log_{17/16}(T))\rceil$, $r_0 \triangleq 2$, *where* $c_0$ *is an absolute constant defined in* (4); *see Appendix B.2. Then for* `Lazy-OPSRL`, *with probability at least* $1 - \delta$,

$$\mathfrak{R}^T = \mathcal{O}\left(\sqrt{H^3SATL^3} + H^3S^2AL^3\right),$$

*where* $L \triangleq \mathcal{O}(\log(HSAT/\delta))$.

*Proof.* Since this proof is very similar to the one of Theorem 3.1 we only describe how it needs to be adapted.

**Optimism** We are going to show that on event $\mathcal{E}^{\mathrm{anticonc}}(\delta)$ (see Proposition B.4 for definition) our estimate of Q-function is optimistic that is $\overline{Q}_h^t(s,a) \geq Q_h^\star(s,a)$ for any $(t,h,s,a) \in \{0,\ldots,T\} \times [H] \times \mathcal{S} \times \mathcal{A}$ and $\overline{V}_h^t(s) \geq V_h^\star(s)$ for $(t,h,s) \in \{-1,\ldots,T\} \times [H] \times \mathcal{S}$.

We prove by forward induction on $t$ and backward induction on $h$. Base of induction $t = -1$ and $h = H+1$ is trivial. Next, if $s \neq s_h^{t+1}$ then $\overline{Q}_h^t(s,a) = \overline{Q}_h^{t-1}(s,a)$ and the statement is correct by the induction hypothesis. In the case of $s = s_h^{t+1}$ we have by Bellman equations and update rule (13)

$$\overline{Q}_h^t(s,a) - Q_h^\star(s,a) = \max_{j\in[J]}\{\widetilde{p}_h^{t,j}\overline{V}_{h+1}^{t-1}(s,a)\} - p_h V_{h+1}^\star(s,a).$$

By induction on $t$ and $h$ we have $\overline{V}_{h+1}^{t-1}(s') \geq V_{h+1}^\star(s')$ for any $s' \in \mathcal{S}$ thus by combination with event $\mathcal{E}^{\mathrm{anticonc}}(\delta)$ we conclude the statement.

**Regret bound** Recall $\delta_h^t = \overline{V}_h^{t-1}(s_h^t) - V_h^{\pi^t}(s_h^t)$ and $\overline{\mathfrak{R}}_h^T = \sum_{t=1}^t \delta_h^t$. By update rule for value function $\overline{V}_h^t(s_h^t) \leq \overline{Q}_h^t(s_h^t, a_h^t)$. Thus we can proceed by update rule for Q-function and Bellman equations

$$\delta_h^t \leq \overline{Q}_h^{t-1}(s_h^t, a_h^t) - Q_h^{\pi^t}(s_h^t, a_h^t) = \max_{j\in[J]}\Big\{\widetilde{p}_h^{t-1,j}\overline{V}_{h+1}^{t-2}(s_h^t, a_h^t)\Big\} - p_h V_{h+1}^{\pi^t}(s_h^t, a_h^t)$$

$$= \underbrace{\max_{j\in[J]}\Big\{\widetilde{p}_h^{t-1,j}\overline{V}_{h+1}^{t-2}(s_h^t, a_h^t)\Big\} - \overline{p}_h^{t-1}\overline{V}_{h+1}^{t-2}(s_h^t, a_h^t)}_{(\mathbf{A})} + \underbrace{[\overline{p}_h^{t-1} - \widehat{p}_h^{t-1}]\overline{V}_{h+1}^{t-2}(s_h^t, a_h^t)}_{(\mathbf{B})}$$

$$+ \underbrace{[\widehat{p}_h^{t-1} - p_h][\overline{V}_{h+1}^{t-2} - V_{h+1}^\star](s_h^t, a_h^t)}_{(\mathbf{C})} + \underbrace{[\widehat{p}_h^{t-1} - p_h]V_{h+1}^\star(s_h^t, a_h^t)}_{(\mathbf{D})}$$

$$+ \underbrace{p_h[\overline{V}_{h+1}^{t-2} - V_{h+1}^{\pi^t}](s_h^t, a_h^t) - [\overline{V}_{h+1}^{t-2} - V_{h+1}^{\pi^t}](s_{h+1}^t)}_{\xi_h^t} + \underbrace{[\overline{V}_{h+1}^{t-2} - \overline{V}_{h+1}^{t-1}](s_{h+1}^t)}_{\Delta_h^t} + \delta_h^t.$$

Here we see that all terms are very similar to the terms that appears in the proof of Lemma B.8 except the additional one $\Delta_h^t \triangleq [\overline{V}_{h+1}^{t-2} - \overline{V}_{h+1}^{t-1}](s_{h+1}^t)$. By adapting the concentration event $\mathcal{G}^{\mathrm{conc}}(\delta)$ with a shift of indices we may obtain the following upper bound (for $N_h^t > 0$)

$$\delta_h^t \leq \left(1 + \frac{1}{H}\right)\delta_h^t + \left(1 + \frac{1}{H}\right)\Delta_h^t + \left(1 + \frac{1}{H}\right)\xi_h^t$$

$$+ 3L\sqrt{\frac{\mathrm{Var}_{\overline{p}_h^{t-1}}[\overline{V}_{h+1}^{t-2}](s_h^t, a_h^t)}{\overline{N}_h^t}} + \sqrt{2L} \cdot \sqrt{\frac{\mathrm{Var}_{p_h}[V_{h+1}^\star](s_h^t, a_h^t)}{N_h^t}}$$

$$+ \frac{10H^2 S \cdot L}{N_h^t} + \frac{16L^2 H}{\overline{N}_h^t}.$$

Thus, the surrogate regret is bounded by almost the same quantity up to a shift of indices and one additional term

$$\overline{\mathfrak{R}}_h^T \leq \tilde{A}_h^T + B_h^T + C_h^T + 4\mathrm{e}H\sqrt{2HTL} + 2\mathrm{e}H^2 SA + \sum_{t=1}^T \sum_{h'=h}^H \gamma_{h'}\Delta_{h'}^t,$$

where $\gamma_h = (1 + 1/H)^{H-h+1}$ and

$$\tilde{A}_h^T = 3\mathrm{e}L \sum_{t=1}^{T} \sum_{h'=h}^{H} \sqrt{\mathrm{Var}_{\overline{p}_{h'}^{t-1}}[\overline{V}_{h+1}^{t-2}](s_{h'}^t, a_{h'}^t) \cdot \frac{\mathbb{1}\{N_{h'}^t > 0\}}{N_{h'}^t}},$$

$$B_h^T = \mathrm{e}\sqrt{2L} \sum_{t=1}^{T} \sum_{h'=h}^{H} \sqrt{\mathrm{Var}_{p_{h'}}[V_{h+1}^\star](s_{h'}^t, a_{h'}^t) \frac{\mathbb{1}\{N_{h'}^t > 0\}}{N_{h'}^t}},$$

$$C_h^T = 26H^2 SL^2 \sum_{t=1}^{T} \sum_{h'=h}^{H} \frac{\mathbb{1}\{N_{h'}^t > 0\}}{N_{h'}^t}.$$

The terms $B_h^T$ and $C_h^T$ remain exactly the same as in the analysis of `OPSRL`, whereas there will be a small difference in the analysis $\tilde{A}_h^T$.

Next, we analyze the new term using non-increasing of the value function $\overline{V}_h^{t-1}(s) \le \overline{V}_h^{t-2}(s)$

$$\sum_{t=1}^{T} \sum_{h'=h}^{H} \gamma_{h'} \Delta_{h'}^t \le \mathrm{e} \sum_{t=1}^{T} \sum_{h'=h}^{H} \Delta_{h'}^t.$$

We derive a bound on the sum of $\Delta_h^t$ over $T$ for any fixed $h$ by a telescoping property

$$\begin{aligned}
\sum_{t=1}^{T} \Delta_h^t &= \sum_{s \in \mathcal{S}} \sum_{t=1}^{T} \mathbb{1}\{s = s_{h+1}^t\}[\overline{V}_{h+1}^{t-2} - \overline{V}_{h+1}^{t-1}](s) \\
&\le \sum_{s \in \mathcal{S}} \sum_{t=1}^{T} [\overline{V}_{h+1}^{t-2} - \overline{V}_{h+1}^{t-1}](s) = \sum_{s \in \mathcal{S}} [\overline{V}_{h+1}^{-1} - \overline{V}_{h+1}^{T-1}](s) \le 2HS.
\end{aligned} \tag{14}$$

Thus we have a next bound for surrogate regret

$$\overline{\mathfrak{R}}_h^T \le \tilde{U}_h^T \triangleq \tilde{A}_h^T + B_h^T + C_h^T + 4\mathrm{e}H\sqrt{2HTL} + 4\mathrm{e}H^2 SA.$$

Next we explain the analysis of term $\tilde{A}_1^T$. To do it, we analyze the sum of variance by following the step of Lemma B.10. All analysis remain exactly the same except the analysis of term $(\mathbf{Z})$, that can be handled by additional use of inequality (14)

$$\begin{aligned}
(\mathbf{Z}) &= \sum_{t=1}^{T} \sum_{h=1}^{H} r_0 H p_h (\overline{V}_{h+1}^{t-2} - V_{h+1}^{\pi^t})(s_h^t, a_h^t) \\
&= 2H \sum_{t=1}^{T} \sum_{h=1}^{H} (\xi_h^t + \delta_h^t + \Delta_h^t) \le 4H^2 \sqrt{2TL} + 2H^2 \tilde{U}_1^T + 2H^2 S.
\end{aligned}$$

The only difference is in the term $2H^2 S$ that is a second-order term. Thus, the following version of Lemma B.10 holds for `Lazy-OPSRL`

$$\sum_{t=1}^{T} \sum_{h=1}^{H} \mathrm{Var}_{\overline{p}_h^{t-1}}[\overline{V}_{h+1}^{t-1}](s_h^t, a_h^t) \mathbb{1}\{N_h^t > 0\} \le 2H^2 T + 2H^2 \tilde{U}_1^T + 40H^3 S^2 AL^3 + 32H^3 S\sqrt{2ATL}$$

with the change only in a constant in front of the third term. The rest of the proof remains the same as in the analysis of `OPSRL`. $\qquad\square$

# G  Experimental details

In this appendix we provides details on comparison OPSRL with some baselines and additionally study the impact of choice of the number of posterior samples $J$ for PSRL and the impact of optimistic prior for OPSRL and PSRL. Our code is published on GitHub and based on the library rlberry by Domingues et al. [2021].

**Environment**    We use a grid-world environment with 100 states $(i, j) \in [10] \times [10]$ and 4 actions (left, right, up and down). The horizon is set to $H = 50$. When taking an action, the agent moves in the corresponding direction with probability $1 - \varepsilon$, and moves to a neighbor state at random with probability $\varepsilon$. The agent starts at position $(1, 1)$. The reward equals to 1 at the state $(10, 10)$ and is zero elsewhere.

**Number of posterior samples**    First we investigate the influence of the number of posterior samples $J$ on the regret. We fixed the other parameters as follows: We use the prior over the transition probability described in Section 3 with $n_0 = 1$ initial pseudo-counts and no inflation $\kappa = 1$. In Figure 2 we plot the regret of OPSRL in the environment described above when the number of posterior samples varies in $J \in \{1, 4, 8, 16, 32\}$. We observe that the number of posterior samples has little effect on the regret, especially if we compare it to the scale of the gap between the different regret curves of the baselines in Figure 1. Thus, in the sequel of this appendix, we arbitrarily choose $J = 8$. Another justification of this choice is that $J \approx \log(T)$ for $T = 10000$, as it was required by theoretical analysis.

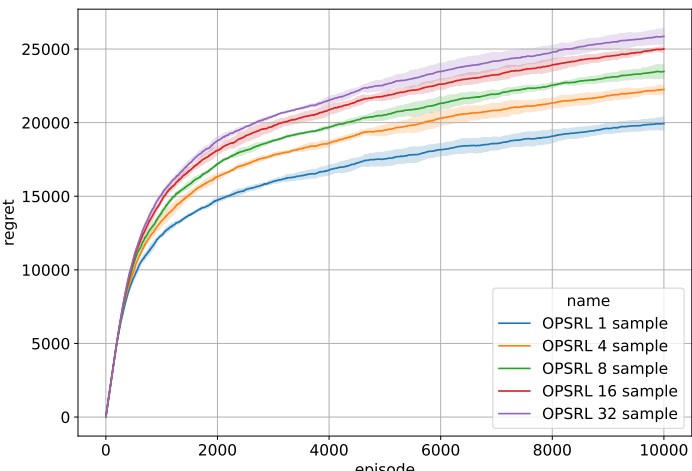

Figure 2: Regret of OPSRL for $J \in \{1, 4, 8, 16, 32\}$ for $H = 50$ and transitions noise 0.2. We show average over 4 seeds.

**Baselines**    We compare OPSRL with the following baselines:

- The UCBVI algorithm by [Azar et al., 2017] (with Hoeffding-type bonuses). Since the theoretical bonus often leads to poor practical performance we use simplified bonuses from an idealized Hoeffding inequality of the form

$$\beta_h^t(s, a) = \min\left( \sqrt{\frac{(H - h + 1)^2}{4 n_h^t(s, a)}}, H - h + 1 \right).$$

- The `UCBVI-B` algorithm, the same algorithm as above but with simplified bonuses from an idealized Bernstein inequality:

$$\beta_h^t(s,a) = \min\left(\sqrt{\frac{\mathrm{Var}_{\widehat{p}^t}[V_{h+1}^{t-1}](s,a)}{n_h^t(s,a)}} + \frac{H-h+1}{n_h^t(s,a)}, H-h+1\right).$$

- The `PSRL` algorithm by [Osband et al., 2013]. For this algorithm we used a Dirichlet distribution of parameter $(1/S, \ldots, 1/S)$ as prior on the transition probability.
- The `RLSVI` algorithm by [Osband et al., 2016b]. As for `UCBVI` we used a simplified variance for the Gaussian noise

$$\sigma_h^t(s,a) = \min\left(\sqrt{\frac{(H-h+1)^2}{4n_h^t(s,a)}}, H-h+1\right).$$

For the `OPSRL` we use the prior over the transition probability described in Section 1 with $n_0 = 1$ initial pseudo-counts and no inflation $\kappa = 1$. Note that the number of pseudo-counts is the same that for the one of the chosen prior for `PSRL` (where the sum of parameters is also one). As discussed above we pick $J = 8$ posterior samples.

**Results**   In Figure 1, we plot the regret of the various baselines and `OPSRL` in the grid world environment. In this experiment, we observe that `OPSRL` achieves competitive results with respect to PSRL. It is not completely surprising since they share the same Bayesian model on the transitions up to the prior. We shall elaborate more on the influence of the prior below. We also note that `OPSRL` outperforms `UCBVI` and `RLSVI`. This difference may be explained by the fact that `OPSRL`'s optimism implies (in the worst case) KL bonuses as in Filippi et al. [2010]. The KL bonuses are stronger than Bernstein bonuses, see Lemma E.1, because they somehow rely on all moments of the empirical distribution rather than the first two moments as in the case of Bernstein bonuses or first moments for Hoeffding bonuses or for the variance of the Gaussian noise in `RLSVI`. Note also that in `OPSRL`, we do not have to solve the complex convex program to compute the KL bonuses Filippi et al. [2010], which could be computationally intensive.

**Influence of prior**   Next we study the influence of the prior for posterior sampling algorithms. Here we will refer to `OPSRL` as an optimistic prior choice and to `PSRL` as a uniform prior choice.

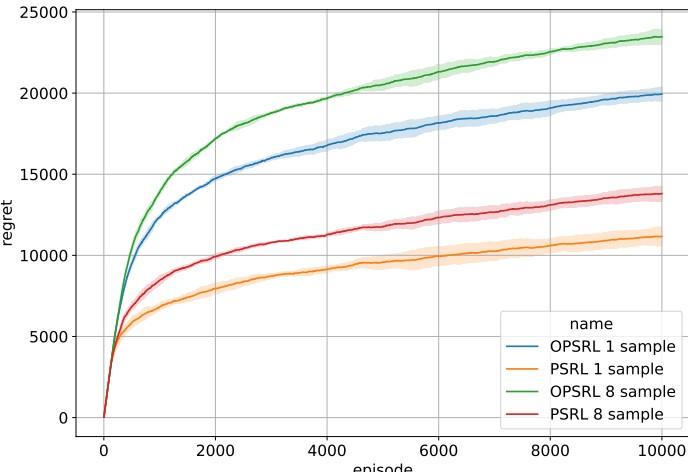

Figure 3: Regret of `OPSRL` with optimistic prior and `PSRL` with uniform prior for $J \in \{1, 8\}$ for $H = 50$ an transitions noise $0.2$. We show average over $4$ seeds.

On Figure 3 we may observe that algorithm convergences for both tested numbers of Thompson samples $J$ and the only difference is the speed of forgetting the prior distribution that results in a

constant difference between regrets. We see that optimistic prior is slightly harder to forget and it is connected to one of the most interesting features of it: optimistic prior is robust to the choice of the underlying probabilistic model. This property makes it universal at the price of efficiency on particular examples.