# OpenReview forum: "Optimistic Posterior Sampling for Reinforcement Learning with Few Samples and Tight Guarantees"
_NeurIPS.cc/2022/Conference — NeurIPS 2022 Accept_

### Official Review · Reviewer_Qhqb · 2022-07-07

**Rating:** 5
**Confidence:** 3
**Soundness:** 2 fair
**Presentation:** 2 fair
**Contribution:** 2 fair

**Summary:**

This paper proposes a novel optimistic posterior sampling algorithm, showing that it can achieve nearly mini-max optimal regret bound with a novel anti-concentration inequality of Dirichlet distribution.


**Questions:**

* I’m not convinced that this method can be better than the frequentist method, as it implicitly mimics the Bernstein type bonus with a sampling method. Can the authors describe what are the potential benefits of the proposed methods over traditional methods like UCBVI? (I admit that UCBVI focuses on the worst-case guarantee and sampling based methods can be better in constant, but I don't think this can lead to e.g. instance-wise benefits.)

* In Line 182-184, is the proposed algorithm applied to general prior other than Dirichlet, as there is only one anti-concentration result for Dirichlet distribution? If not, I think there is no need for this claim.

* Can the proposed methods be generalized to the case with function approximation? It is much more favorable to provide some insights that can be generalized to more general scenarios (e.g. [1]) rather than sticking to the specific tabular setting and Dirichlet distribution.

[1] Zhang, Tong. "Feel-good thompson sampling for contextual bandits and reinforcement learning." SIAM Journal on Mathematics of Data Science 4.2 (2022): 834-857.

**Limitations:**

This is a theoretical paper that does not need to address the potential societal impact.

**Strengths And Weaknesses:**

### Strengths:
* A novel anti-concentration inequality for Dirichlet random vector, which may be of independent interest.

### Weaknesses:
* The application scenario is too limited, as it crucially depends on the Dirichlet distribution. Some artifacts like the pseudo-state $s_0$ may exacerbate this issue.
* No empirical demonstrations of the newly proposed algorithm. The authors claim the benefit of posterior sampling algorithms multiple times (e.g. Line 216-225). Although there are several empirical demonstrations that show that some versions of the optimistic posterior sampling algorithms work better than the empirical counterpart, it’s not clear if such variants can still perform well. Given that this paper is all about tabular settings, I prefer having such an evaluation.
* For me it is much more like an applied math paper, as it does not really solve important questions in reinforcement learning. It mainly show one other potential optimism mechanisms with the anti-concentration of Dirichlet random vector.

---

> ### Author Response · Authors · 2022-08-02
> **Rebuttal 1/2**
>
> We thank reviewer Qhqb for the careful reading and constructive feedback.  Please find below our response to the main points raised in the review.
>
> - The application scenario is too limited, as it crucially depends on the Dirichlet distribution. Some artifacts like the pseudo-state may exacerbate this issue.
>
> We believe that it is important to first better understand a simpler situation of the tabular MDPs before trying to generalize it to a non-tabular case. We would like to emphasize that even the classical PSRL algorithm with the Dirichlet posterior is not yet fully understood and the main goal of our work is to achieve a better understanding of this algorithm from a theoretical point of view.
>
> - No empirical demonstrations of the newly proposed [...], I prefer having such an evaluation.
>
> We designed OPSRL to modify the classical PSRL algorithm as little as possible to maintain empirical performance and at the same time be able to provide a tight regret bound. Actually, this was one of the main motivations of this work. We provide in Appendix G of the rebuttal version an empirical evaluation showing that out of all the other provably optimal algorithms OPSRL achieves the closest performance to classical PSRL. This supports that our modifications did not break the practicality of the original PSRL algorithm.
>
> - For me it is much more like an applied math paper [...] Dirichlet random vector.
>
> This is a philosophical question: should we try to put methods on a sound theoretical basis or propose new methods by validating them experimentally?  We strongly believe that both are very useful: theoretical guarantees allow us to highlight key phenomena, which are often quite subtle, and not easy to grasp by performing only experiments. Theory is a way to foster further methodological developments. We strongly believe that the RL community has benefited greatly in recent years from a better theoretical understanding of algorithms, which provides useful guidance for the development of more sophisticated algorithms. The analysis of RL algorithms today is based on deep probabilistic results, which are indeed more difficult to grasp than the elementary concentration bounds required for the analysis of “simpler” algorithms.
>
> In this paper, we provide the first near-optimal frequentist regret bound for a posterior sampling-like algorithm. This class of algorithms is of great interest to reinforcement learning, as it provides algorithms that perform well empirically [1] and whose exploration mechanism can be easily extended to the DeepRL environment with impressive results [2]. However, little is known theoretically about this class of algorithms. Most theoretical results apply only to the Bayesian framework, with the notable exception of [3], which studies the frequentist framework.
>
> In our work we resolve the following two important open problems stated in [3]:
> Is it possible to propose the variant of posterior sampling algorithm with minimax optimal regret?
> Is it possible to reduce the number of posterior samples for its optimistic counterpart from $\tilde{O}(S)$ to $\tilde{O}(1)$?
> To address these issues, we derived some new concentration bounds for linear forms of Dirichlet random variables. Note that the role of concentration inequalities in the development of efficient reinforcement learning algorithms has already been highlighted in the classical work [1].

---

> > ### Author Response · Authors · 2022-08-02
> > **Rebuttal 1.5/2**
> >
> > - I’m not convinced that this method can be better than the frequentist method, as [...] instance-wise benefits.)
> >
> > We would like to emphasize that our algorithm is more likely to mimic (in the worst case) KL bonuses as in [4]. The KL bonuses are stronger than Bernstein bonuses, see Lemma E.1, because they somehow rely on all moments of the empirical distribution rather than the first two moments as in the case of Bernstein bonuses. In general, however, our algorithm may have a much smaller overestimation error than an optimistic algorithm with any special bonuses. Note also that in OPSRL, we do not have to solve the complex convex program to compute the KL bonuses [4], which could be computationally intensive.
> > Also, we have added an empirical evaluation that shows that the empirical behavior of the posterior sampling methods can be much better than the behavior of the bonus-based classical optimistic algorithms. We also refer the reader to [1] for additional arguments in favor of posterior sampling approaches. The empirical success of the Thompson sampling in stochastic multi-armed bandits [5] already shows that posterior sampling provides a very good exploration mechanism, while UCB with Bernstein bonuses does not perform well empirically in this simple setting (see, for example, the performance of UCB-V in [6]).
> > Finally, we would like to emphasize that the main purpose of our work is to theoretically understand the empirical success of the posterior sampling algorithms in reinforcement learning.
> >
> > - In Line 182-184, is the proposed algorithm [...], I think there is no need for this claim.
> >
> > In the description of the algorithm, we omit the prior to make clear the general principles of our algorithm. In the analysis, we clarify that our results apply to Dirichlet prior/posterior. We would be happy to change the algorithm description if reviewers believe it is misleading.
> >
> > - Can the proposed methods be generalized [...] and Dirichlet distribution.
> >
> > We believe that our algorithm could be generalized to the case with function approximation and think it is an interesting direction of research. We focused on the tabular case since it was already a challenging open problem [3] and a natural first step in developing the core techniques required to analyze the exploration mechanism of a PSRL-like algorithm.
> > But note that there exists already a series of papers that extends PSRL algorithm from tabular to the deep RL setting [7,2,8,9]. Note that these previous approaches only provide Bayesian guarantees for the PSRL regret. On the contrary, our frequentist regret analysis can act as a solid theoretical foundation to further cement the soundness of the PSRL extension to the deep RL setting.
> >
> >  In particular, following the ideas from [9], it is possible to extend the OPSRL algorithm to the deep RL setting as follows. Given transitions $(s,a,h,s\_i)\_{i\in[1,n]}$ for a fixed state-action pair the Q-value at $(s,a)$ is obtained by taking the maximum of Thompson samples of the form $Q\_h (s,a) = r\_h(s,a) + w^n V\_{h+1}(s,a)$ where $w^n \sim \mathrm{Dir}(n\_0,n\_1,\ldots,n\_S)$. Using the aggregation property of the Dirichlet distribution and the fact that it could be obtained by normalizing a vector of independent gamma distributions we can rewrite
> > $ Q_h (s,a)= \sum_{i=-n_0+1}^n z\_i y\_i$ where $y\_i = r\_h(s,a)+V\_{h+1}(s\_i)$ are the targets associated to $(s,a)$ and $z\_i\sim \mathcal{E}(1)$ are independent exponential distributions. Note that we added targets for $i\in[-n\_0+1,0]$ with $s\_i = s\_0$ to take into account the pseudo-transitions.It is then easy to see that one Thompson sample is the solution to a weighted linear regression problem
> > $$
> > Q_h (s,a) \in \mathrm{argmin}\_x \sum\_{i = -n\_0+1}^n z\_i (x-y\_i)^2.
> > $$
> > To obtain the deep RL extension one just needs to solve the above weighted regression with a Q-value function approximated by a neural network. In particular, we recover an alternative version of the bootstrap DQN by [2] where the Bernoulli mask is replaced by an exponential mask and where we take the maximum of the Thompson samples to compute the targets instead of training each Thompson sample with separate targets. Note that, at this point, it is not clear how to deal with the pseudo-transitions but one can add an additive prior as in [8] instead.

---

> > > ### Author Response · Authors · 2022-08-02
> > > **Rebuttal 2/2**
> > >
> > > [1] Osband, I., & Van Roy, B. (2017, July). Why is posterior sampling better than optimism for reinforcement learning?. In International conference on machine learning (pp. 2701-2710). PMLR.
> > >
> > > [2] Osband, I., Blundell, C., Pritzel, A., & Van Roy, B. (2016). Deep exploration via bootstrapped DQN. Advances in neural information processing systems, 29.
> > >
> > > [3] Agrawal, S., & Jia, R. (2017). Optimistic posterior sampling for reinforcement learning: worst-case regret bounds. Advances in Neural Information Processing Systems, 30.
> > >
> > > [4] Filippi, S., Cappé, O., & Garivier, A. (2010, September). Optimism in reinforcement learning and Kullback-Leibler divergence. In 2010 48th Annual Allerton Conference on Communication, Control, and Computing (Allerton) (pp. 115-122). IEEE.
> > >
> > > [5] Chapelle, O., & Li, L. (2011). An empirical evaluation of Thompson sampling. Advances in neural information processing systems, 24.
> > >
> > > [6] Garivier, A., & Cappé, O. (2011, December). The KL-UCB algorithm for bounded stochastic bandits and beyond. In Proceedings of the 24th annual conference on learning theory (pp. 359-376). JMLR Workshop and Conference Proceedings.
> > >
> > > [7] Ian Osband, Daniel Russo, and Benjamin Van Roy. (more) efficient reinforcement learning via posterior sampling. In C. J. C. Burges, L. Bottou, M. Welling, Z. Ghahramani, and K. Q. Weinberger, editors, Advances in Neural Information Processing Systems
> > >
> > > [8] Ian Osband, John Aslanides, and Albin Cassirer. Randomized prior functions for deep reinforcement learning. In S. Bengio, H. Wallach, H. Larochelle, K. Grauman, N. Cesa-Bianchi and R. Garnett, editors, Advances in Neural Information Processing Systems, volume 31. Curran Associates, Inc., 2018.
> > >
> > > [9] Daniil Tiapkin, Denis Belomestny, Eric Moulines, Alexey Naumov, Sergey Samsonov, Yunhao Tang, Michal Valko, Pierre Ménard: From Dirichlet to Rubin: Optimistic Exploration in RL without Bonuses. ICML 2022: 21380-21431.

---

> > > > ### Comment · Reviewer_Qhqb · 2022-08-06
> > > > **Thanks for your response.**
> > > >
> > > > Thanks for the response. I raise the score as the authors provide additional experiments. However, I'm still not convinced that the proposed method can be potentially helpful in the more complicated scenarios, and I hope the authors can continuously work on that.
> > > >
> > > > One minor technical question: To the best of my knowledge, the anti-concentration results in general still focus on certain lower moment conditions (e.g. Gaussian anti-concentration). I'm not sure why the authors claim that the proposed method is more similar to the KL-UCB bonus. Is this a serious claim or is it just some high-level intuition?

---

> > > > > ### Author Response · Authors · 2022-08-08
> > > > > **Post rebuttal**
> > > > >
> > > > > We thank Reviewer Qhqb for reading the rebuttal.
> > > > >
> > > > > The very reason for giving guarantees for OPSRL is actually due to its ability to work in complicated scenarios. Therefore we respectfully disagree with Reviewer Qhqb as we believe that we exhibited in the rebuttal a strong link between the exploration mechanism of OPSRL and the one used by practical algorithms [2,5] in the deep RL setting. This shows the usefulness of our approach in more complicated scenarios than the tabular one.
> > > > >
> > > > > Moreover, we highlight that this last point comes as an additional motivation to the main one: solving the long-standing open problem stated by [3]. Due to its importance, there have been attempts during the past 5 years which were not fruitful and our new technical contributions manage to resolve it.
> > > > >
> > > > > We do claim an explicit link between our way to introduce optimism and the KL bonuses: Indeed at the beginning of the proof of Lemma C.6, around line 790, before reducing to the Bernstein bonuses, we prove that with high probability
> > > > > $$ \max\_{j\in[J]} \tilde{p}^j \bar{V}\_{h+1} \leq \mu$$
> > > > > where $\mu$ is the solution of the following optimization problem
> > > > > $$\mu = \sup \\\{ u : \mathrm{Kinf}(\bar{p},u,\bar{V}\_{h+1}) \leq L \\\}$$
> > > > > where $L = \log(\text{constant}/\delta)/n$. In fact, we can see by using the definition of the $\mathrm{Kinf}$ that $\mu$ is exactly a KL-type bonus:
> > > > > $$\begin{aligned}
> > > > > \mu &= \sup \\\{ u: \exists q \text{ s.t. } q\bar{V}\_{h+1} \geq u \text{ and } \text{KL}(\bar{p},q)\leq L \\\} \\\\
> > > > > &= \sup\\\{ q\bar{V}\_{h+1} :q \text{ s.t. }\text{KL}(\bar{p},q)\leq L \\\}.
> > > > > \end{aligned}$$
> > > > >
> > > > > Thus our UCB that is built as a maximum of Thompson samples is at least as good as the one obtained with KL bonuses.
> > > > > Note that we use a more aggressive exploration function of order $L = \log(\text{constant}/\delta)/n$ instead of the one of order $L = S\log(\text{constant}/\delta)/n$ used by [4] which shaves an extra $\sqrt{S}$ in the final regret bound. It is possible since we control the deviations of $\mathrm{Kinf}(\hat{p},p,V^*\_{h+1})$ instead of $\mathrm{KL}(\hat{p},p)$, i.e we need to control the deviation of the empirical transitions only 'along' the optimal value function.

---

### Official Review · Reviewer_DQbg · 2022-07-11

**Rating:** 7
**Confidence:** 4
**Soundness:** 3 good
**Presentation:** 3 good
**Contribution:** 3 good

**Summary:**

The paper proposed an optimistic posterior sampling algorithm for episodic MDPs and proved that the algorithm has a regret bound matching the lower bound up to poly-log terms.

**Questions:**

As discussed in the paper, the current proof only works when L posterior samples are drawn and L has some logarithmic dependence on H,S,A,T. Given empirical success of posterior sampling with one sample or a fixed number of samples, it would be very interesting if one can have some theoretical guarantees for these settings. From the proof, the size of L seems to come from the union bound on ensuring optimism for all state at all times. Are there some ideas on how the analysis could be extend to settings when L is a constant?

**Limitations:**

The problem setting and limitations are discussed and addressed in the paper.

**Strengths And Weaknesses:**

Strengths:
- The proposed OPSRL algorithm combines ideas from optimism and posterior sampling. Though there are existing algorithms with a similar algorithmic structure (Fonteneau, Korda, Munos, An optimistic posterior sampling strategy for bayesian reinforcement learning, NIPS 2013 Workshop on Bayesian Optimization), OPSRL is the first one with lower bound matching regret guarantee.

- New technical properties on the Gaussian lower bound for Dirichlet distribution is critical in establishing that the sampled Q-function is optimistic, and the exponential upper bound is used in the concentration results. They are useful in proving regret bounds for the episodic MDP setting, and they might be useful for posterior sampling algorithms in other settings as well.


Weaknesses:
- No numerical experience verifying the theoretical results.

---

> ### Author Response · Authors · 2022-08-02
> **Rebuttal**
>
> We thank reviewer DQbg for the careful reading and constructive feedback.  Please find below our response to the main points raised in the review.
>
> - No numerical experience verifying the theoretical results.
>
> We provide preliminary experiments in Appendix G of the rebuttal version of the submission. We compare OPSRL to several baselines. In particular, we show that OPSRL exhibits competitive performance compared to the classical PSRL algorithm and outperforms other algorithms such as UCBVI and RLSVI algorithms.
>
> - As discussed in the paper, the current proof only works when L posterior samples are drawn and L has some logarithmic dependence on H,S,A,T. [...] Are there some ideas on how the analysis could be extended to settings when L is a constant?
>
> Thank you for your interesting question. Unfortunately, it is not clear how to derive a theoretical regret bound for a posterior sampling algorithm with a constant number of samples. Our current result depends heavily on the **union bound argument**, for which we need **a constant probability optimism** for each state-action-step triple in the following sense:
> $$  \tilde{p}^{t,j}\_h V^\star\_{h+1}(s,a) \geq p\_h V^\star\_{h+1}(s,a).$$
> For example, the randomized approach of [1] requires a constant probability optimism not separately for each state-action-step triplet, but a constant probability optimism for almost any convex combination (see Lemma 3 in [1])
> $$\sum d^\pi\_h(s,a) \tilde{p}^{t,j}\_h \overline{V}\_{h+1}(s,a) \geq \sum d^\pi\_h(s,a) p\_h V^\star\_{h+1}(s,a).
> $$
> It is not clear how to extend our anti-concentration result from a linear Dirichlet form to a convex sum of independent linear Dirichlet forms. However, this direction is extremely interesting for further research, as it will explain the empirical success of the classical one-sample posterior sampling algorithm.
>
> [1] Russo, D. (2019). Worst-case regret bounds for exploration via randomized value functions. Advances in Neural Information Processing Systems, 32.

---

> > ### Comment · Reviewer_DQbg · 2022-08-09
> > **Thank you for the response**
> >
> > Thank you for the response. The preliminary experiments are very interesting, and from the experiments it seems like OPSRL with a constant number of samples is performing well. It would be great to extend the current analysis to that direction.

---

### Official Review · Reviewer_uUPZ · 2022-07-11

**Rating:** 6
**Confidence:** 3
**Soundness:** 3 good
**Presentation:** 3 good
**Contribution:** 3 good

**Summary:**

This paper proposed an optimistic variant of the PSRL algorithm and proved a minimax regret bound, which settles an open problem raised by Agrawal and Jia [2017b]. The main technical contribution is a new sharp anti-concentration inequality for Dirichlet random vector, which may be useful for other settings.

**Questions:**

- Can authors elaborate more on how OPSRL adapts automatically to the variance of the estimates so that the optimal dependence on H can be achieved?

**Limitations:**

I am not aware of any potential negative societal impact.

**Strengths And Weaknesses:**

Strengths:
- This paper proved minimax regret bound for an optimistic variant of the widely used PSRL algorithm, which settles an open problem.
- The algorithm is simple and clean.
- The theoretical novelty in developing a tight anti-concentration bound.

Weaknesses:
- OPSRL is very similar to the algorithm analyzed in Agrawal and Jia [2017b] and the main difference is only in applying the new anti-concentration bound, which simplifies some technical ingredients in Agrawal and Jia [2017b].
- OPSRL is still far from the standard PSRL that people use/want to study as multiple posterior sampling is performed at each episode to achieve optimism, which makes the algorithm more similar to UCB algorithm.
- If the authors compare the anti-concentration used in Agrawal and Jia [2017b] with their approach in more detail, it would be easier to understand why the extra S factor is removed.

Minors:
The paper (Randomized Exploration in Reinforcement Learning with General Value Function Approximation) might be relevant as it considers optimistic RLSVI.

---

> ### Author Response · Authors · 2022-08-02
> **Rebuttal**
>
> We would like to thank reviewer uUPZ for the careful reading and the constructive feedback.  Please find below our response to the main points raised in the review.
>
> - OPSRL is very similar [...] ingredients in Agrawal and Jia [2017b].
>
> We have greatly simplified the algorithm of [1] by
>
> 1) removing the simple optimistic sampling part
> 2) providing much better regret guarantees with a novel anti-concentration bound tailored to the Dirichlet-based prior.
>
> An important achievement of our theoretical study is the improvement of the bound $\tilde{O}(H^2 S\sqrt{ AT })$ by [1] to the optimal one $\tilde{O}(\sqrt{H^3 SAT })$. This was attained by removing a spurious $\sqrt{ HS }$ factor.  This in turn allowed us to significantly reduce the number of posterior samples required for the algorithm to achieve the optimal regret bound.
>
> The importance of (anti)-concentration arguments for the development of efficient reinforcement learning algorithms has already been raised by [2] in a very illuminating discussion in Section 4.  Finally, we would like to emphasize that our analysis resolves two long-standing open questions in this area raised by [1], see also [3].
>
>
> - OPSRL is still far from the standard PSRL [...] more similar to UCB algorithm.
>
> We agree that our method is not equivalent to the standard PSRL, but it is the best approximation to the standard PSRL algorithm analyzed so far in a frequentist setting (frequentist regret bounds). The next step in our ongoing research is to provide frequentist regret bounds for PSRL with one sample.
> Moreover, the empirical behavior of our algorithm differs significantly from the behavior of the UCBVI algorithm, as can be seen from the newly added experimental section in Appendix G.
>
>
> - If the authors compare the anti-concentration used in Agrawal and Jia [2017b] with their approach in more detail, it would be easier to understand why the extra S factor is removed.
>
> We have a comparison between our anti-concentration inequality and one by [1] in a note following the sketch of the proof of our anti-concentration inequality. In the revised version of the paper, we highlight this paragraph and add some additional comments. Indeed, our technique for deriving a Gaussian-like lower bound differs substantially from the methodology used by [1]. The latter is based on reducing the distribution of a weighted sum of Dirichlet random vectors to the distribution of a weighted sum of independent beta random variables and a subsequent application of the Berry-Esseen inequality, while our approach is based on the integral representation of the density of the corresponding linear projection of the Dirichlet random vector.
> In particular, the Berry-Esseen inequality is likely to be very coarse since it uses only the first three moments of the distribution and therefore generates an additional S-factor. At the same time, our analysis is much better fitted to the Dirichlet distribution and derives a very tight lower bound. The tightness of our bounds can be checked by comparing it to a similar result for the beta distribution derived in [4].
>
> - Minors: The paper (Randomized Exploration in Reinforcement Learning with General Value Function Approximation) might be relevant as it considers optimistic RLSVI.
>
> We would thank the reviewer for the additional reference. We will add it to the literature review.
>
> - Can authors elaborate more on how OPSRL adapts automatically to the variance of the estimates so that the optimal dependence on H can be achieved?
>
>
> The way of adaptation to the variance could be observed by directly computing the variance of our estimate
> $$\mathrm{Var}[ \tilde{p}^{t,j}_h(s,a) \bar{V}^t_h(s,a) ] =\frac{\mathrm{Var}\_{\bar{p}^{t}_h}(s,a) [\bar V^t_h(s,a)]}{\alpha+1}.
> $$
> Another way to observe this is to note that the optimism we introduce leads to sharper upper confidence bounds than those that can be derived using Bernstein-type bonuses (see the Bernstein-type inequality presented in Lemma 3.5 or Lemma C.6 in Appendix C). This shows that our algorithm is as good as UCBVI with Bernstein bonuses in the worst case. And it is well known that variance-aware bonuses are the key to achieving a tight regret bound in $H$.
>
> [1] Agrawal, S., & Jia, R. (2017). Optimistic posterior sampling for reinforcement learning: worst-case regret bounds. Advances in Neural Information Processing Systems, 30.
>
> [2] Osband, I., & Van Roy, B. (2017, July). Why is posterior sampling better than optimism for reinforcement learning?. In International conference on machine learning (pp. 2701-2710). PMLR..
>
> [3] Russo, D. (2019). Worst-case regret bounds for exploration via randomized value functions. Advances in Neural Information Processing Systems, 32.
>
> [4] Alfers, D., & Dinges, H. (1984). A normal approximation for beta and gamma tail probabilities. Zeitschrift für Wahrscheinlichkeitstheorie und verwandte Gebiete, 65(3), 399-420.

---

> > ### Comment · Reviewer_uUPZ · 2022-08-06
> > **Thanks for your reply**
> >
> > Thanks for your reply! After reading other reviews and replies, I still think this paper is an interesting contribution to the community and would like to keep my score.

---

### Comment · Area_Chair_pJA3 · 2022-08-05
**@Reviewers, please read response and engage in discussions!**

Please acknowledge that you have read the rebuttal.

---

### Meta-Review · Area_Chair_pJA3 · 2022-09-08

**Recommendation:** Accept
**Confidence:** Certain

**Metareview:**

The paper propose a variant of posterior sampling RL, namely OPSRL, and provided a minimax regret analysis for this algorithm. This settles an open question in the RL regret theory. The key technical novelty is a new anti-concentration inequality, which can be used to improved the analysis by Agrawal and Jia 17 and closed their gap. Reviewers appreciate this theoretical contribution. Some reviewer questioned the applicability of the method and lack of numerical experiments. The authors supplied experiments after rebuttal, which largely addressed some of these questions. One reviewer raised his score from reject to borderline acceptance. Overall, this is a solid paper with meaningful theoretical contribution to posterior sampling and to RL theory.

**Award:**

No

---

### Decision · Program_Chairs · 2022-09-14

Accept